# Trends and variability in the Southern Annular Mode over the Common Era

Jonathan King [1,2] ✉, Kevin J. Anchukaitis [1,2,3], Kathryn Allen [4,5,6], Tessa Vance[7] & Amy Hessl [8]

The Southern Annular Mode (SAM) is the leading mode of atmospheric variability in the extratropical Southern Hemisphere and has wide ranging effects on ecosystems and societies. Despite the SAM's importance, paleoclimate reconstructions disagree on its variability and trends over the Common Era, which may be linked to variability in SAM teleconnections and the influence of specific proxies. Here, we use data assimilation with a multi-model prior to reconstruct the SAM over the last 2000 years using temperature and drought-sensitive climate proxies. Our method does not assume a stationary relationship between the SAM and the proxy records and allows us to identify critical paleoclimate records and quantify reconstruction uncertainty through time. We find no evidence for a forced response in SAM variability prior to the 20th century. We do find the modern positive trend falls outside the $2\sigma$ range of the prior 2000 years at multidecadal time scales, supporting the inference that the SAM's positive trend over the last several decades is a response to anthropogenic climate change.

The Southern Annular Mode (SAM) is the leading mode of atmospheric variability in the extratropical Southern Hemisphere and is characterized by a mostly zonally-symmetric mass oscillation with anti-correlated pressure anomalies over the mid-latitudes and Antarctica [Refs. 1–4, and see Fig. 1]. The SAM's phases capture the strength and position of the mid-latitude westerly winds and the subtropical jet, such that positive phases promote a poleward shift of storm tracks and intensification of the circumpolar westerly belt, while negative phases promote an equator-ward shift of storm tracks and weakening of the westerly winds. Variability in the SAM therefore has wide ranging effects across the Southern Hemisphere. Positive phases of the SAM are linked to cooling over Australia and central Antarctica, as well as warming over the Antarctic Peninsula and southern South America[5–11]. Hydroclimate effects of the positive phase include drying over southern South America, western South Africa, southern Australia, and

New Zealand, as well as increased precipitation over central and eastern Australia and southeastern South America[7,10,12–16]. SAM variations have also been linked to wildfire activity in South America and south-east Australia[17–21], changes in sea ice distribution[8,22–28], temperature anomalies over East Antarctica[9,29,30], ice shelf collapse[31,32], and ocean-atmosphere carbon exchange[33–35]. Understanding SAM variability is therefore important for both societies and ecosystems throughout the Southern Hemisphere, particularly in sub-tropical-temperate regions projected to experience a future drying climate.

Since the 1950s, the SAM has exhibited a trend toward a more positive state[4,36,37], which has been attributed to stratospheric ozone depletion and rising concentrations of atmospheric $CO_2$[38–42]. This positive trend has potentially contributed to severe droughts, including the Day Zero Cape Town drought[43] and Millennium Drought in Australia[44,45], as well as increased fire activity[17–19]. Given these impacts,

[1]Department of Geosciences, University of Arizona, Tucson, AZ 85721, USA. [2]Laboratory of Tree-Ring Research, University of Arizona, Tucson, AZ 85721, USA. [3]School of Geography, Development, and Environment, University of Arizona, Tucson, AZ 85721, USA. [4]School of Geography, Planning and Spatial Sciences, University of Tasmania, Hobart 7001, Australia. [5]School of Ecosystem and Forest Sciences, University of Melbourne, Richmond, VIC 3121, Australia. [6]Centre of Excellence for Australian Biodiversity and Heritage, University of New South Wales, Sydney, Australia. [7]Australian Antarctic Program Partnership, Institute for Marine and Antarctic Studies, University of Tasmania, Hobart, Australia. [8]Department of Geology and Geography, West Virginia University, Morgantown, WV, USA. ✉e-mail: jonking93@arizona.edu

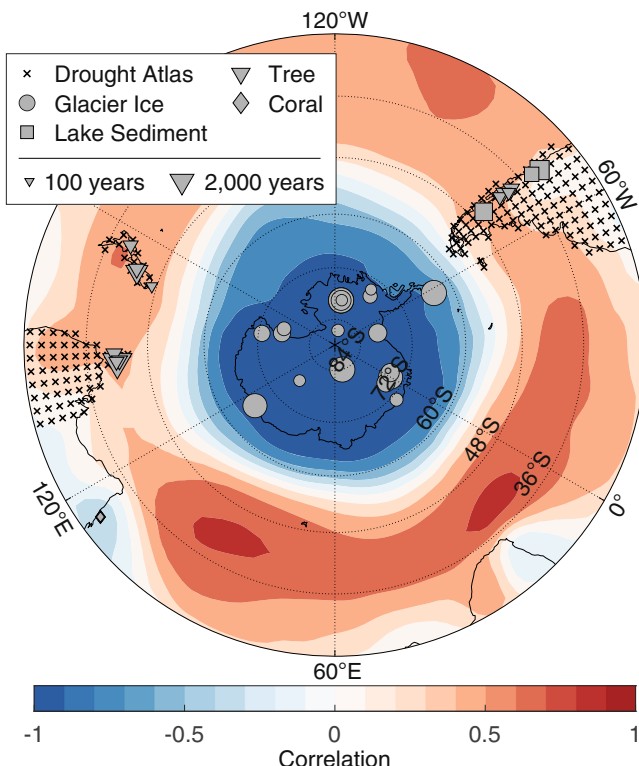

**Fig. 1 | Proxy network map.** Black Xs indicate the centroid of binned drought atlas sites. Gray markers indicate PAGES2k sites. The size of the PAGES2k markers correspond to the length of each record. Filled color contours show the field correlation between the Southern Annular Mode index and austral summer (December–February) mean sea level pressure from the 20th Century Reanalysis[117,118] over the period 1958-2000 CE.

it is important to place the SAM's recent behavior in a long-term perspective and assess the relative influence of anthropogenic forcing and natural climate variability. In the context of multi-decadal trends, reconstructions spanning multiple centuries are necessary to resolve forced responses from the SAM's internal variability. Instrumental records of the SAM only extend through the mid-1900s and longer reanalysis-derived indices show low correlations with one another and differences in variability prior to the 1950s[37,46], so characterizing the SAM's long-term behavior requires paleoclimate reconstructions derived from natural climate archives.

There are several existing multi-century SAM reconstructions[47–49] (henceforth, V12, A14, and D18), but they show limited agreement prior to the 1850s[49,50]. Most indicate a negative phase in the SAM during the late 1400s, but there is a marked lack of decadal to centennial coherence between the reconstructions prior to the 1800s[50]. There are several potential reasons for these differences. Firstly, all three reconstructions rely on the calibration of proxy records directly with an instrumental SAM index. This implicitly makes two important assumptions for each reconstruction: first, that the relationship of proxy records to local climate variables is stationary over time; and second, that the SAM's teleconnections with local climate variables are stationary and well-represented by the instrumental record. While the first is reasonable and a necessary assumption of most paleoclimate analyses, multiple studies cast doubt on this second point, and regional complexity in the climate response to specific SAM phases further decreases the likelihood of this assumption holding. For instance, even over the instrumental period, SAM exhibits non-stationary connections with precipitation and temperature anomalies in southern South America, Australasia, and the Antarctic Peninsula[13,51], and many of the proxy records in existing SAM reconstructions come from these areas[50].

Rising concentrations of greenhouse gases, changes in stratospheric ozone, connections with ENSO, spatial changes to the SAM's structure, and stochastic climate variability can also affect the SAM's influence on regional climates over multi-decadal time scales[36,39,52–54]. Pseudo-proxy experiments have also shown that non-stationary teleconnections cause reconstruction skill to vary widely with the selection of different calibration windows[55]. This effect is particularly pronounced for proxy networks with fewer than 20 sites, which is common in the early portions of SAM reconstructions. To mitigate such effects, D18 explicitly screened for stationarity in their reconstruction, although this required calibration with a longer and therefore less reliable observational record[56].

Differences between SAM reconstructions may also result from the selection of different reconstruction targets and proxy networks. For example, A14 targets an annual SAM index, whereas V12 and D18 target an austral summer (DJF) SAM index. D18 found that annual reconstructions were much more sensitive to the selection of proxy sites and calibration windows and they conclude that annual products may exhibit increased sensitivity to non-stationary teleconnections, which may partly explain the differences between the reconstructions. Additionally, each index has been reconstructed using a different proxy network with a different geographic extent. A14 targets the Drake Passage sector, using a mix of terrestrial proxy types from southern South America as well as Antarctic ice cores. In comparison, V12 targets the Pacific sector, using a network of tree-ring chronologies from South America and New Zealand. D18 uses the most spatially extensive network, including tree-ring records[47], Antarctic ice cores, PAGES2k South American proxies[57], and coral records from the tropical Pacific[58]. Furthermore, A14 utilizes a temperature-sensitive proxy network, while V12 and D18 leverage both temperature and hydroclimate-sensitive proxies. Given the variability of the SAM's teleconnections on regional scales[13,51], and the climate sensitivities of different proxy types[56], the variations in proxy-network design may further help explain reconstruction differences. It is often difficult to assess the influence and contribution of individual proxy records in multiproxy reconstructions, so the cause of any reconstructed index's behavior are often unclear. This is particularly relevant in the period prior to 1400 CE, when the sparsity of proxy networks leaves the reconstructions vulnerable to the dominant influence of just a few records. Ultimately, as a consequence of these uncertainties and the differences in existing reconstructions, the evolution of the SAM over the Common Era and its response to external forcing remains poorly constrained[50,59].

To address these uncertainties, here we reconstruct the austral summer (DJF) SAM index over the Common Era at annual resolution using offline paleoclimate data assimilation (DA). As a reconstruction technique, paleoclimate DA integrates climate proxy records with the dynamical behavior captured by climate models[60,61]. In brief, most paleoclimate DA approaches use forward or proxy-system models[62,63] to translate climate model states into the same dimensions or space as a collection of climate proxy records. This allows direct comparison of the model output with the proxy records. The climate model states are then updated to more closely match the proxy records, and a model-derived estimate of climate system covariance is used to propagate the update to reconstruction targets, such as the SAM. DA has been used to reconstruct surface air temperature anomalies[64–67], geopotential height fields[65], the response to volcanic eruptions[68,69], sea ice extent[70,71], sea surface temperatures[72,73], sea level pressure and winds[70,74], and hydroclimate variables[75]. Leveraging reconstructions of atmospheric circulation, two of these studies have already used DA to reconstruct the SAM[70,74] (hereafter D21 and O21). However, these reconstructions are thus far limited to the last one to two hundred years, and so do not provide detail on the SAM's response to external forcings prior to the 1800s. Here, we reconstruct the SAM over the full Common Era. To do so, we use an offline ensemble Kalman filter to

assimilate 40 records from the PAGES2k temperature-sensitive proxy network[57], the South American Drought Atlas (SADA)[76], and the Australia-New Zealand Drought Atlas (ANZDA)[77] (Fig. 1). Our method uses a stationary multi-model ensemble constructed from over 4,500 years of output from four general circulation climate models. We rely on linear seasonal-temperature forward models for the PAGES2k network, and we use non-linear Palmer Drought Severity Index (PDSI) estimators for SADA and ANZDA. Here, we follow Gong and Wang[2] and define the SAM index using the difference of zonal-mean pressure anomalies (see Methods, Eq. (1)).

In the context of SAM paleoclimate reconstructions, DA offers several additional advantages relative to traditional methods. Firstly, our method does not calibrate proxy records against a SAM index, and so does not assume stationary SAM teleconnections. Instead, the DA relies on the calibration of proxy forward models to local temperature and precipitation near the proxy sites, which only assumes the stability of proxy relationships to their local climate. Additionally, we estimate covariance between proxies and the SAM using thousands of years of climate model output. As a result of this, our proxy-SAM relationships are not sensitive to potentially anomalous decadal- or centennial-scale variations in the SAM's behavior. Furthermore, DA is amenable to the use of a range of proxy types as well as gridded climate records with spatial autocorrelation, and we leverage this to incorporate the two existing tree-ring based drought atlases into our reconstruction. Previous work indicates that SAM reconstructions using hydroclimate-sensitive sites are more skillful than those using strictly temperature-sensitive proxy networks[55]. Each drought atlas provides extensive coverage for at least the last five centuries and each incorporates over 150 tree-ring records. They therefore represent a significant source of hydroclimate information available for our reconstruction.

Finally, our DA method allows us to incorporate an optimal sensor analysis[78] as part of the final reconstruction. Traditionally, optimal sensor analyses have been used to identify ideal regions for future proxy development[78–81]; however, they can also be applied within a DA framework to quantitatively assess the power or value of different proxy sites as the overall network evolves through time. We use this to identify the proxy sites that are most likely to influence the reconstruction in each time step, which helps characterize the reconstruction's overall behavior. This information is particularly useful in the early part of this Common Era reconstruction, when the sparse network size can give high weights to a limited number of records.

## Results

### Reconstruction

Because we use a stationary prior, the temporal variability of the raw reconstruction depends on the size and composition of the proxy network. As the proxy network becomes sparse, less information is incorporated in the Kalman Filter, and the updated state is less able to move away from the prior mean. This causes reconstruction variability to increase with the size of the proxy network and independently of the climatological record. We apply a variance correction scheme to account for this effect (Supplementary Fig. 4). Variance adjustments are common in paleoclimate reconstructions[82–85] and are inherent to simpler methods like Composite Plus Scale[86] in order to avoid misinterpreting variance changes due to sample size or methodology as reflecting true climatic causes. We refer to the variance-corrected reconstruction in all following discussion.

We next assess the skill of our SAM reconstruction relative to the Marshall[4] and Fogt indices[87,88], two commonly used instrumental SAM indices (see Methods for further details). Before comparing time series, we first normalize the Fogt index and our reconstruction to the Marshall index, such that the mean and variance of the detrended normalized time series match those of the detrended Marshall index over the period 1958-2000 CE. This places all series in the same unit space while preserving differences in the instrumental trend.

**Table 1 | Reconstruction Skill metrics for the Southern Annular Mode calculated against instrumental indices over the given time periods**

| Metric | Marshall index (1958-2000) | Fogt index (1958-2000) | Fogt index (1866-2000) |
|---|---|---|---|
| Correlation ($p \ll 0.001$) | 0.72 | 0.67 | 0.56 |
| RMSE | 1.45 | 1.56 | 1.80 |
| σ Ratio | 0.97 | 1.03 | 1.15 |
| Mean Bias | −0.26 | 0.45 | −0.29 |

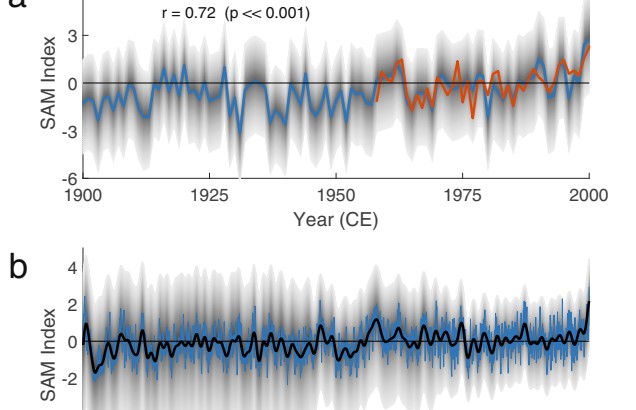

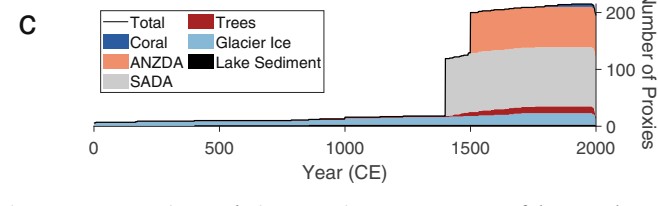

**Fig. 2 | Reconstruction evolution over time. a** Comparison of the annual reconstruction (blue) with the Marshall index (red) over the instrumental era. Shading indicates the 5–95 percentiles of the reconstruction. **b** Evolution of the annual reconstruction (blue) and 31-year lowpass filtered (black) over the Common Era. Shading indicates the 5–95 percentiles of the lowpass filtered series. **c** Composition of the proxy network over time. Colors for proxy types are as follows: Dark blue (coral), pink (Australia–New Zealand Drought Atlas), gray (South American Drought Atlas), dark red (trees), light blue (glacier ice), black (lake sediment).

Examining skill values (Table 1; Fig. 2a), we find that the reconstruction's correlation with the Marshall index (1958-2000 CE) is $r = 0.72$ ($p \ll 0.001$), which is comparable to that reported for A14 ($r = 0.75, p \ll 0.001$)), and somewhat higher than those of D21 ($r = 0.30, p < 0.05$) and O21 ($r = 0.35 - 0.37, p < 0.05$). With respect to the 20th century Fogt index, our reconstruction correlates at $r = 0.65$ ($p \ll 0.001$), somewhat higher than A14 ($r = 0.51, p \ll 0.001$)). All p-values reported for our reconstruction account for temporal autocorrelation[89]. Our RMSE values with the Marshall index (1.45) are similar to, albeit slightly higher than, those reported by D18 (1.32). As discussed in the introduction, our reconstruction is not calibrated to a SAM index, and so we emphasize that the agreement with the Marshall and Fogt indices is not built-in to our reconstruction method and thus represents an independent skill metric. We also note that the definitions of the Marshall and Fogt indices differ from the Gong and Wang index used by our reconstruction; thus, we would not expect perfect agreement even for the most skillful reconstruction.

We next characterize the reconstruction's behavior over the last two millennia (Fig. 2b). However, we first note that the sign of a given

anomaly is in part a function of the normalization period. Thus, a change in sign indicates a relative shift in the strength and position of the mid-latitude westerly winds and subtropical jet, but is not associated with a specific physical meaning. The reconstruction exhibits minimal evidence for trends over most of the first millennium of the Common Era, although the third and seventh centuries are both marked by increased multidecadal variability as the SAM alternates between negative and positive phases. A strongly negative anomaly in the early 1000s is followed by a notable 100-year positive trend that concludes with the most positive anomalies outside of the instrumental era. The SAM persists in a positive state until the late 1400s, when it abruptly decreases to strongly negative values. After this event, the index returns to near-zero mean anomalies. It has a peak in the mid-1700s and begins exhibiting a positive trend in the early 1800s. This trend intensifies in the later half of the 20th century, and the reconstruction ends with the most positive SAM anomalies observed during the Common Era.

Reconstruction uncertainty ranges from ±4.5 anomaly units in the early reconstruction to less than 2.3 after 1500 CE (Fig. 2b). We note that, because we use a stationary prior, the reconstruction years are treated as fully independent of one another. While this is common in many reconstruction techniques, it does not represent the reality of the SAM, which exhibits persistence on interannual time scales due to potential connections with the stratosphere[90,91], tropical variability[92,93], and external forcing[42,94]. Because our paleoclimate DA design does not incorporate inter-annual persistence, the uncertainty estimates shown here likely overestimate the true reconstruction uncertainty. Overall, uncertainty decreases as the reconstruction approaches the present day, a result of the increasing size of the proxy network (Fig. 2c).

### Optimal sensor
We use our optimal sensor framework to identify which proxies are most responsible for reducing reconstruction uncertainty over time (Fig. 3). The potential for a proxy to reduce uncertainty corresponds to the proxy's influence on the reconstruction, so this analysis also allows us to identify which proxies most strongly influence the reconstruction at a given point in time. The first 900 years of the reconstruction are most strongly affected by the temperature-sensitive Mt. Read (Tasmania) tree-ring record with additional support from the Plateau Remote, WDC06A, and WDC05A ice cores. At 900 CE, the temperature-sensitive Oroko (New Zealand) tree ring chronology joins the network and supplants Mt. Read as the most influential record. Two large decreases in reconstruction uncertainty occur in 1400 and 1500 CE, which correspond to the addition of the SADA and ANZDA, respectively.

### External forcing
We next examine the reconstruction's response to external forcing and find little evidence for response to external forcings in the pre-industrial period. Figure 4a displays a wavelet coherence sample plot between the reconstruction and the solar forcing time series. Coherence between the two would be observed as a statistically significant horizontal band, and given the lack of such a band, we detect no evidence for a link between our reconstruction and solar forcing. We next use superposed epoch analysis to examine the response to volcanic forcing. Values outside of the gray bands in Fig. 4b would indicate a statistically significant and consistent response to volcanic forcing. Since the composite-mean response remains within these bands, we again detect no evidence for a consistent response to volcanic forcing in our reconstruction, potentially because the magnitude of unforced variability overwhelms any volcanic signal.

By contrast, the reconstruction exhibits significant positive trends in the latter half of the twentieth century. Figure 4c, d shows calculated trends in the Marshall Index and the reconstruction over the instrumental period. Each plot illustrates values for trends of different

lengths, centered on different sets of years. Here we have normalized trends in the Marshall Index and reconstruction to a similar unit space (see Methods). To test the significance of these trends, we use the reconstruction to establish the distribution of trends over the intervals 1-1900 CE and 1500–1900 CE. For each interval, we determine the distribution of trends of each length. For each trend length, we use the 95% confidence interval (CI) of the corresponding distribution to establish a significance threshold. In other words, modern trends outside of these thresholds are significantly different at the $2\sigma$ level from the trends in the intervals used to establish the natural distributions.

We find that the reconstruction exhibits significant positive trends in the latter half of the twentieth century. However, these modern trends are only significant on time-scales greater than approximately 40 years, and trends over shorter time scales fall within the reconstructed 95% CI of possible trends from natural variability. Examining the Marshall index, we similarly find that trends shorter than about 35 years are within the reconstructed 95% CI of natural variability, but that trends longer than about 35 years fall outside this range. The Marshall index exhibits its most positive, significant trends for intervals centered on the early 1980s. Although this period is near the end of our reconstruction and less well resolved than preceding decades, we note that the reconstruction similarly exhibits strongly-positive, significant trends centered on the early 1980s. Here we have quantified natural variability using the distribution of reconstructed trends over the period 1500–1900 CE, the years including both drought atlases. If we instead use the period of the full reconstruction (1–1900 CE), the tests become more stringent. Statistically significant trends in the reconstruction are limited to the last 60-80 year interval, and Marshall index trends are only significant when containing the interval 1964–2000 CE. We also experimented with using the early portion of the reconstruction (1–899 CE) to quantify natural variability (Supplementary Fig. 1). We find these results are qualitatively similar to those using the full reconstruction period, in that the most recent long-term trends are outside the range of natural variability. A notable period where similar persistent multidecadal trends as those observed over the recent period are identified in the reconstruction is in the middle of the 11th century (Fig. 5). However, uncertainties during this period are substantial due to the lack of proxy data, particularly the drought atlases. D18 also shows a similar feature at this time, and therefore this period would benefit from additional proxy data and analytical scrutiny. Radiative forcing is not remarkable during this period[95] and there is no significance coherence with solar forcing (Fig. 4a). Existing large-scale temperature reconstructions also disagree during this period[96].

## Discussion
Our reconstruction suggests that the SAM is dominated by internal variability throughout the pre-industrial Common Era. This finding is in agreement with D18, who likewise found minimal influence of solar and volcanic forcing on their reconstruction. Volcanic signals have likewise been a challenge to detect in Southern Hemisphere temperature reconstructions[97]. Some studies have proposed that an observed relationship between SAM and ENSO[48,93,98–100] could provide a pathway for solar forcing[101] to influence the SAM[50,102]; however, our results do not support this mechanism during the the Common Era. In a set of model simulations, Wright et al.[102] found that increasing the amplitude of the prescribed solar variability lead to a significant relationship between solar forcing and the simulated SAM. These authors suggest that using high amplitude solar forcing could help reconcile SAM reconstructions with climate simulations; however, the lack of solar signals in our reconstruction differs notably from their findings and instead further supports the realism of low-amplitude solar forcing scenarios[103–105].

By contrast with solar and volcanic forcings, our analysis indicates that the most recent multi-decadal trend is outside the 95% CI of

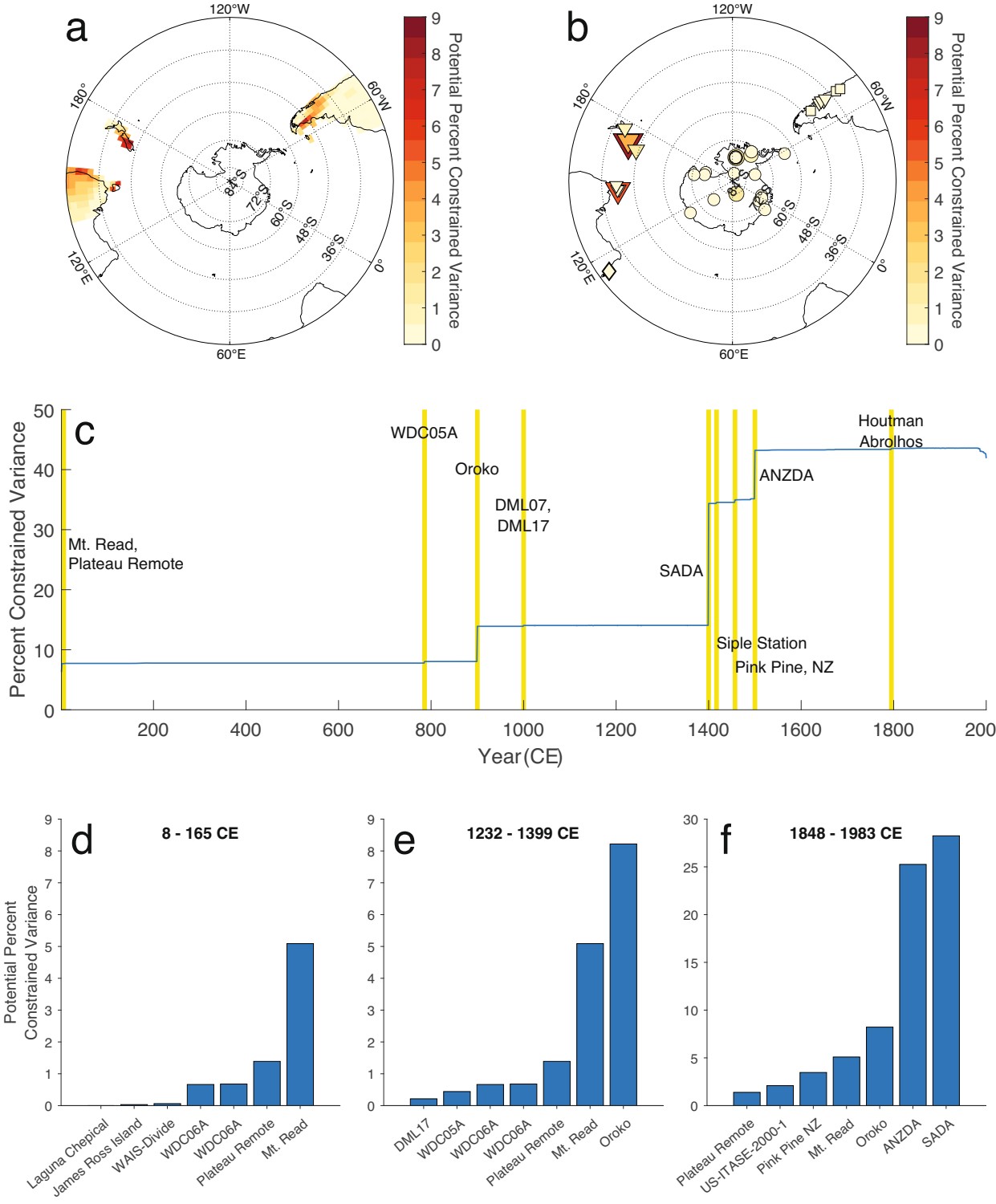

**Fig. 3 | Optimal sensor analysis. a, b** Maps of the potential ability for drought atlas (**a**) and PAGES2k sites (**b**) to constrain reconstruction posterior variance (red shading). **c** Evolution of reconstruction posterior variance over time. Yellow bars indicate the addition of the indicated proxy to the network. **d–f** Ranked histograms of the seven sites with greatest potential influence in the early reconstruction (**d**), immediately before the addition of drought atlases (**e**) and for the full network (**f**). Potential influence is determined as the uncertainty constrained by a single-proxy network.

natural variability and is consistent with the SAM's response to anthropogenic forcing. We emphasize that this modern trend is only significant for intervals longer than about 40 years when assessed against the 1500–1900 CE period, or intervals of about 55 years when considering the full Common Era. Shorter trend periods remain within the 95% CI of natural variability, even for the most recent intervals. The

significance of the modern positive trend therefore reflects its anomalous persistence, rather than the amplitude of its decadal-scale variation alone. The significance of these longer trends emphasizes the importance of the paleoclimate record, particularly given the uncertainties in instrumental SAM records prior to the late twentieth century[46,106]. We also note that the modern positive trend is only

**Fig. 4 | Climate forcing analysis. a** Wavelet coherence of the reconstructed Southern Annular Mode (SAM) index with the solar forcing reconstruction[95]. Black contours surround statistically significant coherencies. Strong coherence with the solar forcing series would appear as a horizontal band of contours. **b** Composite mean response of the SAM in years preceding and following major volcanic events (blue line). Shading indicates the range of natural variability at the 95% confidence level assessed from the reconstruction. Composite mean values outside of the shading would suggest a consistent, statistically significant response to volcanic forcing. **c** Instrumental trends for the Marshall index. Colored squares indicate trend values calculated over different periods. Values are calculated from a window centered on the year denoted on the X-axis. The length of trend is given by the duration on the Y-axis. Solid contours surround trends that are significantly different (at the 95% confidence level) from the natural distribution of trends in the reconstruction over the period 1500–1900 CE. Dotted contours show results using the period 1–1900 CE to assess significance. **d** As in (**c**), but for trends calculated from the reconstruction. The units of the reconstruction (and thereby its trends) have been normalized to the Marshall index, such that mean and variance of the detrended reconstruction match those of the detrended Marshall index over the interval 1958–2000 CE.

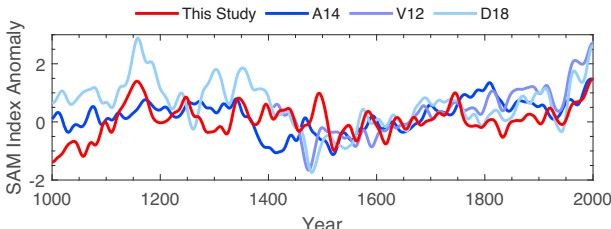

**Fig. 5 | Reconstruction comparison.** Comparison of Southern Annular Mode reconstructions over the last millennium. Previous reconstructions include A14[48], V12[47], and D18[49]. All reconstructions are smoothed via a 30-year Gaussian filter and normalized to the period 1400–1850 CE.

outside of the 95% CI of natural variability for trends spanning the years from about 1940–2000 CE. Trends are generally not significant during the early 1900s, and are even negative for the 50 year period centered on the 1930s. These results help establish the onset of the modern positive trend at around 1940 CE. This timing coincides with increasing emissions of ozone-depleting substances, such as chlorofluorocarbons, and greenhouse gases, and is consistent with literature attributing the modern trend to stratospheric ozone depletion and rising levels of atmospheric $CO_2$[38–42].

We next compare our reconstruction with the V12, A14, and D18 products (Fig. 5). We normalize the mean and variance of each index over the period 1400–1850 CE to allow comparison of the series in the same unit space. We select the year 1400 CE because it is the first year with values for all four reconstructions and we end the normalization in 1850 CE to limit the sensitivity of our comparison to differing representations of the post-industrial trend. All four indices agree on the existence of a strong positive trend during the late twentieth century; however, all show limited coherence with one another prior to about 1850 CE, as noted in previous studies[49,50]. Ultimately, the limited agreement of these reconstructions reduces confidence in the significance of modern trends[59]. As discussed in the introduction, the potential causes of these discrepancies include differing seasonal targets, different proxy networks, and the relative weights of proxies

within those networks[50]. For instance, temperature sensitive tree-rings in the Andes do not show much influence in our optimal sensor analysis (Fig. 3), even though these are an important part of the V12 reconstruction. Similarly, the sedimentary pigment record from Lake Aculeo that is a significant predictor in A14 does not have a large influence in our DA reconstruction. Additionally, V12, A14, and D18 rely on calibration with the instrumental SAM index, which can cause uncertainty when there is non-stationarity in the teleconnection of local climate with the SAM. Ultimately, one new reconstruction does not solve the problem of differing reconstructions and ours similarly shows limited agreement with all of V12, A14, and D18. However, our assimilation is not calibrated to the SAM, and instead relies on calibrating proxy forward models to local climate variables. As such, the assimilation offers a potential improvement by reducing uncertainty arising from non-stationary SAM teleconnections.

We also compare our reconstruction with D21 and O21. To better compare the three DA products, we compute correlation coefficients and $p$-values for D21 and O21 following the methods outlined for our own reconstruction. The D21 reconstruction correlates with the Marshall index at a value of $r = 0.30$, $p < 0.05$. O21 provides four reconstructions, each using a different climate model as the prior. From O21, we find the PACE and HadCM3 reconstructions have $p > 0.05$, but the remaining CESM and LENS reconstructions correlate at $r = 0.37$, $p < 0.05$ and $r = 0.35$, $p < 0.05$, respectively. Both D21 and O21 correlate with the Marshall index at lower levels than our reconstruction, despite also using DA. D21 uses an offline particle filter, whereas our assimilation and O21 rely on an offline Kalman filter. Thus, methodological differences may in part contribute to differences between the reconstructions. Additionally, both D21 and the individual O21 reconstructions rely on priors derived from a single climate model. By contrast, our prior uses a multi-model ensemble, which can help reduce the effects of individual climate model biases on an assimilation and thereby improve skill[107]. Finally, differences in our proxy networks may also influence the different reconstructions. In particular, our use of drought atlases represents a significant source of hydroclimate information not available in the D21 and O21 reconstructions and likely contributes to reconstruction skill over the last several centuries (Fig. 3c, f).

An additional advantage of our reconstruction is the transparency provided by the optimal sensor's assessment of the relative weights and influence of proxy records in our network. In general, we find that our reconstruction is most strongly influenced by the two drought atlases, followed by the Mt. Read (Tasmania), Oroko (New Zealand), and Pink Pine (New Zealand) tree ring chronologies, and also the Plateau Remote, Siple Station, WDC06A, and WDC06B ice cores. We note here that a minor change in reconstruction uncertainty does not imply that a proxy has a weak effect on the reconstruction, because highly influential proxies from the same location may present redundant climate signals. For example, the Pink Pine chronology is the third most potentially influential PAGES2k record (Fig. 3f), but has a relatively small effect on reconstruction uncertainty when added to the network in 1457 CE (Fig. 3c). This is because much of the Pink Pine climate signal is already represented by the nearby Oroko site. However, such redundant sites are valuable because they make the reconstruction less sensitive to non-climatic noise from a single highly-influential proxy record. In the case of Pink Pine and Oroko, spreading the southern New Zealand climate signal over two influential records allows either site to partially account for non-climatic noise in the other. A proxy's potential influence reflects both its covariance with the SAM and the ability of our proxy estimates to accurately estimate the record. Ultimately, assuming our estimates of climate covariance are accurate, the influential sites are those most likely to contribute skill to the reconstruction.

Overall, we find that tree-ring chronologies from Tasmania and New Zealand, the West Antarctic ice cores, and the drought atlas

locations in Tasmania, southern New Zealand, the eastern edge of Australia, and southeast South America all have the greatest potential for reconstructing SAM (Fig. 3a, b). This suggests that additional proxy development in these regions, or extensions of shorter existing records such as the Oroko and Pink Pine tree-ring chronologies or the Siple Station ice core, would be valuable for improving the skill of future SAM reconstructions. However, we caution that location alone is not sufficient for proxy utility and recommend that future proxy development should demonstrate a robust sensitivity to local climate that reliably connects proxy records to the SAM. In our optimal sensor framework, a proxy's potential influence is a function of (1) the accuracy of our proxy forward models, and (2) the covariance of the resulting proxy estimates with the SAM in the climate models. As a result, our analysis may currently undervalue proxies from regions with limited climate model agreement, and future improvements in both climate and proxy system models may allow paleoclimate data from other regions to contribute to skillful reconstructions of the SAM.

Our DA method does not require a calibration with the instrumental SAM, which helps limit sensitivity to non-stationarity in the SAM during the instrumental era. However, the trade-off is the influence of proxy forward model and climate model biases on the reconstruction. In the case of proxy models, any biases typically reduce the weight of the proxy in the assimilation, thereby limiting its effect on the reconstruction. Improving the accuracy or sophistication of the proxy forward models could increase the influence of many records; for example, transitioning the statistical forward models used here for the PAGES2k sites to more mechanistically accurate proxy system models[62] could potentially improve the reconstruction[108]. However, efforts to develop more complex proxy system models must also exercise caution, as excessive complexity and poorly constrained parameters may lead to overfitting and artificially high skill in the instrumental era at the expense of accuracy during the earlier reconstruction. In this study, we retain the simpler statistical forward models because (1) the PAGES2k proxies are reported to be temperature sensitive[57], (2) statistical proxy models remain the most common and tractable approach for paleoclimate data assimilation to date[61,65,67], and (3) the simple statistical model eliminates errors caused by the interaction of climate model biases with forward models that rely on absolute units.

With respect to climate models, biases in the mean state can affect proxy estimates that include parametrizations or thresholds based on absolute units. However, covariance biases are a greater concern, as they introduce errors in the propagation of information from the proxy records to the reconstruction target. For example, some of the climate models considered in this study simulate a SAM pattern that is too zonally symmetric and that overestimates the SAM's influence on overall Southern Hemisphere circulation[109]. Such teleconnection biases can cause the assimilation to overestimate the covariance between various proxies and the SAM, thereby increasing reconstruction error. In this study, we use a multi-model ensemble (MME) to reduce the effects of covariance bias from any one model[67,107]. We note that we weight each model equally, which effectively treats each model as independent. In reality, many models share common features or code, so this equal weighting may bias an ensemble towards the most similar models[110,111]. For example, the CCSM4 and CESM-LME output used in our MME are both from models developed by the US National Center for Atmospheric Research (NCAR) and may more closely resemble one another than the MPI or MRI models. Future efforts may wish to test different model composition or weights when constructing a MME prior.

Finally, our use of a stationary offline prior implies a stationary estimate of climate system covariance when considered over the full reconstruction period. Although we use a long-term estimate of the SAM's climate covariance, the true covariance may vary on multi-decadal scales[13,51], and these variations will not be captured in our

approach. While the assumption of a reasonably stationary covariance is implicitly common to most spatial reconstruction methods[112,113], the application of transient offline priors[73,114,115] or online assimilation techniques[116] may enhance future data assimilation reconstruction, although these approaches must balance the utility of evolving covariance estimates with reduced ensemble sizes.

Our study provides the first reconstruction of the Southern Annular Mode at annual resolution over the entire Common Era. We use a data assimilation method that does not calibrate the proxies directly against the instrumental SAM index, so the reconstruction is not sensitive to known SAM non-stationarity in the modern era. Our reconstruction leverages both the SADA and ANZDA in addition to the PAGES2k proxy network and represents a significant increase in paleoclimate information available to reconstruct the SAM. Optimal sensor analysis indicates that the first 1400 years of the reconstruction are strongly influenced by the Oroko and Mt. Read tree-ring chronologies, with additional support from the Plateau Remote, WDC06A, and WDC05A ice cores. As the SADA and ANZDA are added to the proxy network (1400 CE and 1500 CE, respectively), the drought atlases become strong drivers of the reconstruction's behavior.

Our reconstruction provides a foundation with which to assess the drivers of the SAM's behavior over the Common Era. Such assessments are critical given the SAM's importance to societies and effects on climate variability throughout the Southern Hemisphere. Prior to the most recent decades, we find no relationship between the SAM's variability and natural external climate forcing, suggesting that the SAM's behavior is dominated by internal unforced variability over the pre-industrial Common Era. We then examine the recent positive trend in the SAM, which is linked to increased drought severity, wildfire intensification, reduced sea ice distribution, and Antarctic ice shelf collapse. We find that this trend is outside the 95% confidence interval of natural variability for the last millennium, further indicating the modern positive trend is likely a response to anthropogenic forcing.

## Methods

### Southern Annular Mode Index

In this study, we use the Gong and Wang[2] definition of the SAM index:

$$SAM = P^*_{40°S} - P^*_{65°S} \qquad (1)$$

where $P^*_X$ indicates the normalized zonal-mean sea level pressure (SLP) at a particular latitude. The latitudes 40°S and 65°S were selected as the zonal-means with the most strongly anti-correlated SLP anomalies across the mid- and high-latitude Southern Hemisphere. We use this definition, as opposed to an index derived from a principal component analysis, because the latitudes of the most strongly anti-correlated SLP anomalies are robust across the climate models considered in our assimilation (Supplementary Tables 1, 2). We target the austral summer (DJF; December-February) SAM because this corresponds to the seasonality of the climate response of the majority of our proxy network. D18 also suggests that summer SAM reconstructions are more robust to proxy network design than annual reconstructions, which further supports this choice. When calculating the SAM index, we normalize seasonal mean values, rather than individual months. Austral summers span months from two calendar years, and this can introduce date ambiguities for annual records, particularly tree-ring chronologies. Throughout this paper, we use the convention that the year of an austral summer value matches the calendar year of the associated January.

### Reanalysis and Instrumental Indices

We use monthly precipitation and air-temperature fields from the Twentieth Century Reanalysis V3 (20CR)[117,118] to calibrate our DA method. The 20CR is based on an 80-member ensemble Kalman Filter, and extends from 1850 CE to present at 2 degree resolution. Because of

its role in our assimilation method, this effectively sets an upper bound on the resolution of any gridded spatial product used in this reconstruction. We also use the austral summer Marshall index[4] and Fogt index[87,88] to assess the skill of our reconstruction in the modern era. The Marshall index estimates the Gong and Wang[2] definition of the SAM (Eq. 1), and is based on data from 12 weather stations (6 near 40°S, and 6 near 65°S). Because it uses station data, the Marshall index is not subject to the spurious trends observed in high-latitude Southern Hemisphere reanalysis pressure fields[4]. The Fogt index is constructed using a principal component regression of station pressure data and calibrated to the Marshall index. These indices are commonly used as a comparison point for SAM reconstructions[47–49].

### Climate proxies

In this reconstruction, we assimilate the PAGES2k temperature-sensitive proxy network[57], the South American Drought Atlas (SADA)[76], and the Australia-New Zealand Drought Atlas (ANZDA)[77]. We limit all three datasets to those sites or locations south of 25°S. Pseudo-proxy tests of other latitude bounds suggests that reconstruction skill is minimally affected by the use of more northward proxy sites and agreement with the instrumental record exhibits a slight maximum for a bound at 25°S (Supplementary Fig. 2). Overall, this domain maximizes the number of SAM-sensitive proxy sites in our network, while minimizing the effects of distal proxies that primarily reflect other climate signals.

From the PAGES2k dataset, we include all sites from the PAGES2k global temperature reconstruction that have annual or sub-annual temporal resolution. To maintain a common timescale, we bin all sub-annual sites to annual resolution. Our PAGES2k network therefore consists of 40 proxy records: 12 tree-ring chronologies, 3 lake sediment cores, 5 corals, 19 ice-cores, and 1 borehole-derived temperature reconstruction (Supplementary Table 3). The tree-ring records are from Tasmania, New Zealand, and the central Andes. The longest two chronologies are from Mt. Read, Tasmania and Oroko, New Zealand, which begin in 494 BCE and 900 CE, respectively; the remaining tree chronologies mostly begin between 1450 CE and 1550 CE. The three lake sediment proxies are derived from the central and southern Andes. The longest record (Laguna Chepical) spans the complete Common Era, while Lagunas Escondida and Aculeo begin in 400 CE and 816 CE, respectively. The five coral records are from the Houtman Abrolhos Islands off the west coast of Australia and begin between 1795 CE and 1900 CE. The Antarctica ice core records have varying temporal coverage. Four sites cover the full Common Era (Plateau Remote, WDC06A, James Ross Island, WAIS-Divide), six more extend at least one millennium, and the remaining nine begin between 1140 CE and 1703 CE. The borehole reconstruction is from WAIS-Divide and begins in 8 CE. For the 40 proxy set, full coverage extends from 1903 CE to 1983 CE with 20 sites remaining by 2000 CE.

The SADA and ANZDA are gridded tree-ring reconstructions of the self-calibrated Palmer Drought Severity Index (PDSI) during austral summer at annual resolution[76,77]. The SADA is derived from 286 temperature and precipitation-sensitive tree-ring chronologies and begins in 1400 CE. The atlas covers all of South America south of 12.25°S at 0.5° resolution. Similarly, ANZDA is derived from 176 tree-ring chronologies, as well as one coral record, and begins in 1500 CE. The ANZDA covers Australia east of 136.25°E, and New Zealand, also at 0.5° resolution. The SAM is strongly associated with droughts and pluvials in the domains of both atlases[76], supporting their inclusion in our network. Both atlases have significantly higher spatial resolution than the reanalysis data and climate model output used for our reconstruction method. To permit calculations that require the same spatial resolution, we bin both atlases to the lowest resolution spatial grid relevant to a given experiment. For the main reconstruction, after applying latitude screening, our SADA and ANZDA networks consist of 104 and 71 binned records, each on a 2° x 2.5° grid. It is worth noting that

several of the PAGES2k tree ring records used in our reconstruction were also used to construct the drought atlases, and these repeat records might initially appear to duplicate information in the reconstruction. However, our Kalman filter method explicitly accounts for covariance between proxy records, and down-weights proxies with repeated information accordingly. Additional details for this process can be found in the following section.

## Kalman Filter

Our reconstruction uses an ensemble Kalman Filter approach (EnKF)[119], which follows the update equation:

$$\mathbf{X_a} = \mathbf{X_p} - \mathbf{K}(\mathbf{Y} - \hat{\mathbf{Y}}) \qquad (2)$$

in each reconstructed time step. Here, the $\mathbf{X_p}$ and $\mathbf{X_a}$ matrices are the initial (prior) and updated (analysis) ensembles of climate model states. Each row holds a target climate variable, and each column a different selection of climate model output (ensemble member). $\mathbf{Y}$ is a matrix of proxy values for the time step; the columns of $\mathbf{Y}$ are constant, and each row holds the value from a particular proxy record repeated once for each ensemble member. $\hat{\mathbf{Y}}$ holds the model estimates of the proxy values; each row has the estimates for a particular proxy site, and each column has the estimates from a particular ensemble member. $\mathbf{K}$ is the Kalman gain:

$$\mathbf{K} = \mathrm{cov}(\mathbf{X_p}, \hat{\mathbf{Y}})[\mathrm{cov}(\hat{\mathbf{Y}}) + \mathbf{R}]^{-1} \qquad (3)$$

where $\mathbf{R}$ is the matrix of proxy error-covariances. The Kalman filter accounts for duplication of information across repeated proxy records. This occurs via the $\mathrm{cov}(\hat{\mathbf{Y}})$ term in Eq. (3), which reduces proxy weights in the Kalman gain as a function of shared proxy covariance. Note that any shared covariance derived from proxies' relationships with the SAM is balanced by the $\mathrm{cov}(\mathbf{X_p}, \hat{\mathbf{Y}})$ term in Eq. (3). We use a square-root variant of EnKF[120,121]. This modifies Eqs. (2) and (3) to update the ensemble mean and deviations separately, and precludes the need for perturbed observations[122]. The Kalman filter can be expressed as a recursive Bayesian filter[123,124], so we will often refer to $\mathbf{X_p}$ and $\mathbf{X_a}$ as the prior and posterior in this paper.

## Prior

We construct the prior using output from climate models with paleoclimate simulations of the last millennium (Supplementary Table 1). We use a multi-model ensemble (MME), which has been found to reduce error relative to single model assimilations[67,107]. Our MME consists of CCSM4, CESM-LME, MPI, and MRI, which represent the set of last millennium simulations with spatial resolutions greater than or at the resolution of the 20CR reanalysis. As such, this selection does not require us to bin the drought atlases to lower resolutions than 20CR, which allows us to extract maximum information from SADA and ANZDA. We also tested a larger MME consisting of 10 models with last millennium simulations regardless of resolution. Our tests show that the high-resolution MME maximizes reconstruction skill (Supplementary Fig. 3). For CCSM4, MPI, and MRI, we use output from the PMIP3 last1000 (850–1850 CE) and historical (1851–2005 CE) experiments, specifically ensemble member r1i1p1. For CESM-LME, we use output from full-forcing run 2 (850–2005 CE). While the PAGES2k proxy network does include stable oxygen isotope proxies, there are too few high-resolution last millennium isotope-enabled paleoclimate model simulations available to construct a multi-model prior[107].

We use an offline, stationary prior for our assimilation. Offline approaches[125,126] differ from classical Kalman Filters in that updates are not used to inform model simulations. Instead, offline methods use pre-existing model output to build the prior in each time step. The offline approach has been shown to compare favorably with classical (online) methods in paleoclimate contexts but at a fraction of the

computational cost[127,128]. The stationary prior indicates that we use the same ensemble as the prior for each reconstructed time step. This is common in paleoclimate DA applications[60,65,108] and is justified by the limited forecast skill of climate models beyond the annual reconstruction time scale[114]. However, stationary priors have been observed to artificially reduce the variability of reconstructions as proxy networks become more sparse[67]. Consequently, our use of stationary priors necessitates a correction for the reconstruction's variability, which is detailed in the methods below.

To build each prior, we first calculate the DJF SAM time-series for each model, normalizing zonal SLP means to the pre-industrial period (850-1849 CE). We then concatenate the SAM index time-series from each model in every year of model output. The final prior has a total of 4624 ensemble members from 4 high-resolution models.

## Proxy forward models and error covariances

The proxy modeling process begins by designing a forward model for each assimilated proxy record. Here, we use different forward models for the PAGES2k and drought atlas products. For the PAGES2k records, we follow previous studies[61,67] and use simple univariate linear models:

$$\hat{\mathbf{Y}} = a\mathbf{T} + b \qquad (4)$$

where $\hat{\mathbf{Y}}$ is a vector of proxy estimates, and T is a vector of seasonal temperature means. Here, the seasonal mean used for each site is taken from the seasonal sensitivity reported in the PAGES2k metadata[57]. We determine the coefficients $a$ and $b$ by calibrating each proxy PAGES2k record to the corresponding climate data from 20CR. For each proxy site, we first determine the seasonal sensitivity and then linearly regress the proxy record against the seasonal-mean temperature vector from the closest 20CR grid point in all overlapping years from 1950–2000 CE. The regression slope and intercept are then used as $a$ and $b$, respectively. For the drought atlases, we estimate proxies by calculating PDSI[129] using the Thornthwaite estimation of potential evapotranspiration[130]. This uses monthly mean temperature and precipitation from a drought atlas grid cell to compute monthly PDSI values for each year. We then use the austral summer means of these monthly values as the proxy estimates. Effectively:

$$\hat{\mathbf{Y}} = \mathrm{mean}[\mathrm{PDSI}_{\mathrm{Thornthwaite}}(\mathbf{T}, \mathbf{P})]_{\mathrm{DJF}} \qquad (5)$$

where $\mathbf{T}$ and $\mathbf{P}$ are monthly temperature and precipitation, and $\hat{\mathbf{Y}}$ is the drought atlas estimate. We estimate proxy values for the model priors by applying Eqs. (4) and (5) to climate model output and matching each year's estimates to the associated ensemble member in the prior.

Although the PDSI calculation in Eq. (5) uses the Thornthwaite approximation, both drought atlases target an observational dataset based on the Penman–Monteith method[76,77,131]. However, both the Thornthwaite and Penman–Monteith equations have been shown to perform similarly when applied to pre-industrial simulations, and this agreement occurs because the simplifying assumptions of the Thornthwaite method remain valid over the relatively confined range of last millennium temperatures[132]. For the purposes of this study, the Thornthwaite method provides two further advantages: First, the Thornthwaite equation is more computationally tractable, which allows us to apply it to the large spatial regions and the multiple millennium-length climate model simulations used for priors in our assimilation method. Second, because the Thornthwaite calculation requires fewer climate model data fields to estimate the PDSI[130,131], opportunities for climate model biases to degrade the reconstruction are reduced.

We next estimate the proxy error covariances. These error covariances describe the uncertainty in the comparison of observed records to the proxy estimates ($\mathbf{Y} - \hat{\mathbf{Y}}$). In a classical Kalman Filter, the estimates ($\hat{\mathbf{Y}}$) are known perfectly and this uncertainty is derived from

the observations (**Y**), so **R** is often referred to as observation uncertainty. In paleoclimate contexts, this situation is inverted: proxy measurements are typically precise and uncertainty derives from the simplifications and parameterizations inherent in the estimation equations. Hence, we quantify R by running Eqs. (4) and (5) on the 20CR dataset (from 1950-2000 CE) and comparing the estimated proxy values to the real records. The differences between the two sets of values are used to estimate the errors inherent in using simple models and relatively coarse climate data to estimate the temporal behavior of the proxy records. Most EnKF paleoclimate efforts assume that proxy errors are independent, such that **R** is a diagonal matrix[60,61,65,75]. This is justified for datasets like PAGES2k, for which proxy uncertainties are dominated by local biological, physical, and mechanistic effects[56]. However, the drought atlas grid points are strongly spatially correlated, so this assumption is not appropriate in this study. Instead, we calculate independent error-variances for the proxies in the PAGES2k network, and full error-covariances for both SADA and ANZDA. Hence, **R** is block-diagonal, rather than strictly diagonal. We estimate uncertainty in the final reconstruction from the spread of the assimilation posterior.

### Variance correction

We use a series of frozen-network assimilations to adjust the temporal variance of the reconstructed SAM index. There are five sites in our proxy network with observations in every year of the reconstruction. We first assimilate this five-site network over the full interval 1–2000 CE to derive a baseline time-series that is not affected by changes to the proxy network. We next determine each unique set of proxy sites used to update one or more time steps in the reconstruction. We then assimilate each set of proxies over the time steps for which all the proxies in the set have recorded values, and we determine the ratio of this assimilation's standard deviation to that of the baseline time series over all overlapping years:

$$P(\text{set}) = \sigma_{\text{set}} / \sigma_{\text{Baseline}} \qquad (6)$$

We then calculate a scaling factor for each time step using the normalized ratio for the associated proxy set:

$$w(t) = P(\text{set}(t)) / \max(P) \qquad (7)$$

A comparison of the raw and variance-adjusted reconstructions is provided in Supplementary Fig. 4.

### Optimal sensor analysis

Our optimal sensor analysis uses a Kalman filter framework to estimate the ability of proxy sites to reduce the variance of a metric across a posterior ensemble[78]. The reduction of variance for the kth proxy site is given by:

$$\Delta\sigma_k = \text{cov}(\hat{\mathbf{Y}}_k, \mathbf{J})^2 [\text{var}(\hat{\mathbf{Y}}_k) + \mathbf{R}_k]^{-1} \qquad (8)$$

where **J** is the metric, $\hat{\mathbf{Y}}_k$ are the proxy estimates for the site, and $\mathbf{R}_k$ is the site's error-variance. Here we use the SAM index as our metric, so the optimal sensor analysis assesses the ability of sites to reduce uncertainty in the SAM index across the reconstruction posterior. We first compute the total reduction in SAM posterior variance using the complete set of proxies with observations in each time step. We also quantify each site's ability to reduce reconstruction uncertainty when no other sites are in the proxy network. We refer to this quantity as potential percent constrained variance.

### External forcing analyses

We begin our external forcing analysis by investigating the SAM's response to natural climate forcings. We first use a wavelet coherence analysis to examine the relationship between our SAM reconstruction and a time series of reconstructed solar forcing[85,95]. We next use a superposed epoch analysis (SEA)[133] to determine the reconstruction's composite mean response to major volcanic eruptions. We used the eVolv2k V3 volcanic forcing dataset[134,135] to select events with a total forcing magnitude greater than or equal to that of Krakatoa. This yielded 28 eruption years: 87, 169, 266, 433, 536, 540, 574, 626, 682, 817, 939, 1108, 1171, 1182, 1230, 1257, 1276, 1286, 1345, 1458, 1600, 1640, 1695, 1783, 1809, 1815, 1831, and 1883. For the SEA, we normalized each event to the mean of the preceding 5 years and examined the composite mean response over the 10 years following volcanic events. We tested the significance of the observed response by bootstrapping 5000 SEA time series via random draws of 28 event years from the remaining years in the reconstruction.

We next consider the SAM's response to anthropogenic forcings using both our reconstruction and the Marshall index. Before quantifying trends, we first normalize our reconstruction to the Marshall index, such that the mean and variance of the detrended normalized reconstruction match those of the detrended Marshall index over the years of common overlap (1958–2000 CE). This places the series in the same unit space while preserving differences in the instrumental trend. We then calculate moving trends for the reconstruction over the years 1900–2000 CE using trend window lengths from 31 to 101 years. Similarly, we calculate moving trends for the Marshall index over the years 1958-2020 CE using trend window lengths from 31 to 63 years. We then use the reconstruction to assess the significance of these trends. For each trend window length, we calculate the distribution of trends with the given window length from the reconstruction over the years 1500–1900 CE, and we define this distribution as the natural variability for that trend length. We then use the 95% confidence intervals of each distribution to determine a significance threshold for trends of the associated length. We also repeat this process using trend distributions from the intervals 1–1900 CE and 1–899 CE to examine the sensitivity of this analysis to different portions of the reconstruction.

## Data availability

The final reconstructed SAM index time series, its uncertainty estimates, and the analyses generated in this study have been deposited in a Zenodo repository at https://doi.org/10.5281/zenodo.7643732. The input climate model datasets required to implement the analysis are available via the Earth System Grid Federation https://esgf-node.llnl.gov/projects/esgf-llnl/ and the NCAR Climate Data Gateway https://www.earthsystemgrid.org/.

## Code availability

The code required to reproduce our analyses, figures, and tables are available in Zenodo repository https://doi.org/10.5281/zenodo.7643732. The repository includes all third-party packages used by our code, but these are also available externally. Specifically, the DASH toolbox is available at https://github.com/JonKing93/DASH, the PDSI model is also available at https://github.com/JonKing93/pdsi, and the so was wavelet analysis package is available at https://tocsy.pik-potsdam.de/wavelets/.

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

## Acknowledgements

The authors thank David Meko for providing the original PDSI estimation code on which ours is based. This research is supported by a grant from the US National Science Foundation's Paleo Perspectives on Climatic Change program (P2C2) AGS-1803946 to K.J.A. and A.H; an Australian Research Council Special Research Initiative for Antarctic Gateway Partnership (SR140300001) to T.V.; a grant from the Australian Antarctic Program Partnership (ASCI000002) to T.V., and a grant from the Australian Research Council (FT200100102) to K.A. We acknowledge the World Climate Research Programme's Working Group on Coupled Modelling, which is responsible for CMIP, and we thank the climate modeling groups for producing and making available their model output. We also acknowledge the many proxy paleoclimatologists whose work has contributed to the large-scale networks and databases used in this paper.

## Author contributions

K.J.A., A.H., and J.K. designed the research. J.K. wrote all the software and code, improved and enhanced the assimilation method, conducted all calculations, and produced all the figures. A.H., T.V., and K.A. provided guidance, data, and knowledge for individual proxies and Southern Hemisphere climate variability. All authors (J.K., K.J.A., A.H., T.V. and K.A.) interpreted the results and wrote the paper.

## Competing interests

The authors declare no competing interest.
