## [Peer Review File · Nature Communications]

Trends and variability in the Southern Annular Mode over the Common EraReviewer #1 (Remarks to the Author):

Review for Authors, King et al., Trends and variability in the Southern 2 Annular Mode over the Common Era

King and colleagues have produced the first data-assimilation based reconstruction of the SAM over the Common Era. The reconstruction shows a positive trend over the 20th century, which the authors attribute to anthropogenic forcing. The DA reconstruction is compared to three others published over the past several years, and also is used to investigate which proxies contribute the most information to the reconstructed climate variability. This information is useful in the context of a sensor placement optimization exercise, and may inform further proxy development in the critically under-studied Southern Hemisphere.

This is a well-written paper worthy of publication in NComms for its novelty and findings, but I do have some comments that could strengthen the presentation and clarity of the results and conclusions:

General Comments:

1. I was craving more information about the proxy system models used in the main text. This can be very confusing, because the use of that phrase has very different meaning in the DA literature compared to the physically-based proxy system modeling literature (e.g. Evans, Dee et al.) – are the authors truly using a PSM, like VS Lite, or just a linear or bi-variate linear calibration with temperature and precipitation? At least one sentence being clear about this in the earlier sections of the text would be helpful.

2. The use of multiple model priors is great, and should be emphasized in the main text.

3. It is fundamentally interesting that the reconstructions differ so much before the 20th century. I'm wondering if, beyond the speculative text from 271-288, if the authors could find a way to truly diagnose the drivers of the divergence in the reconstructions? Or prove / quantify the effects of "seasonal expressions" for example?

4. In the abstract and elsewhere, the authors spend time discussing how their reconstruction circumvents issues central to other studies like stationarity, but I feel these ideas are underdeveloped.

5. The paper would benefit in the conclusions but perhaps also at the end of the abstract with. A statement on how the positive trend in the SAM might manifest in terms of climate changes, or compound with Southern Hemisphere climate change? For example, in which regions does the positive trend in the SAM cause drought, and what are the implications for Antarctic sea ice?

Specific Comments:

25 variability and trends over what timescale?

30 this seems like a bit high-level for the abstract, and without context doesn't make sense here – the idea of stationarity between proxy records and SAM, perhaps phrase this more globally or move it to the discussion/results../ intro

34 implications of your findings? Last sentence should be implications – perhaps remove the sentence mentioned above at 30 and replace with this.

53-55 what about Antarctic sea ice, ice shelf collapse, mitigation of warming around E. Antarctica, climate change compounding impacts?

72 I'm still wary of this very broad use of the words "trends and variability" – in what fields, and on what time scales, and how do these features indicate "large discrepancies?"

123 Not exactly recent, 2016 is already 6 years ago and the original Gosse papers came out well before that!

138 This sentence seems out of place and perhaps belongs in the methods, or in a different section.

179-180 I'm a little confused by this on first read, can you be more explicit about what you mean when you say "our reconstruction is not calibrated directly to the SAM index." You are reconstructing the SAM index,....

202 why? (why does the connection to stratosphere/tropical variability etc.) lead you to say the uncertainty is overestimated?

206 sentence structure here personifies "proxy" a bit too much for my style preferences, I would say "the potential for any one proxy record to reduce ..."

217 use of "common" here is a bit vague, do you mean consistent?

218 Figure 4 is quite difficult to interpret without more information in the main text.

233 rephrase? = "A significant trend in the ..."

235 rephrase "We also experimented"

236 similar how?

268 ozone-depleting substances – I would spell out what those are like CFCs etc. and be specific

320-321 I was craving more information about the proxy system models used in the main text. This can be very confusing, because the use of that phrase has very different meaning in the DA literature compared to the physically-based proxy system modeling literature (e.g. Evans, Dee et al.) – are the authors truly using a PSM, like VS Lite, or just a linear or bi-variate linear calibration with temperature and precipitation? At least one sentence being clear about this in the earlier sections of the text would be helpful.

Based on Section starting in methods 489, it appears you are simply using linear regression.

But the main text suggests dependencies on both moisture and temperature (?)

340 suggest starting new sentence "Such assessments.."

341 might be worth mentioning the explicit effects, eg. Drought, fire, etc. here again – this is important.

342 it is fundamentally interesting that the reconstructions differ so much before the 20th century. I'm wondering if, beyond the speculative text from 271-288, if the authors could find a way to truly diagnose the drivers of the divergence in the reconstructions? Or prove / quantify the effects of "seasonal expressions" for example?

349- can you close with a statement on how the positive trend in the SAM might manifest in terms of climate changes, or compound with Southern Hemisphere climate change? For example, in which regions does the positive trend in the SAM cause drought, and what are the implications for Antarctic sea ice?

Supplemental Figure 1, what are the units of the trend??

Supplemental Table S3 the total number of proxy records used in the reconstruction should be mentioned in the main text. Also Figure 2 doesn't seem to jive with Table S3, perhaps because of the addition of the SADA and ANZDA data?

All figures should be labeled with panels e.g. a, b, c, d for the caption clarity.

Figure 4 caption/main text needs more explanation on how to read these figures.

Reviewer #2 (Remarks to the Author):

The paper is well-written in a clear and logical style. The methods and materials section is exhaustive and should enable the work to be reproduced. The noteworthy component of the paper is that it uses a new (to this field) proxy data assimilation technique to compute a 2000-year proxy index of the Southern Annular Mode (SAM). This is an important but relatively new area of research in Southern Hemisphere climatological studies. Previous SAM reconstructions of similar length are quite different from each other, which significantly reduces the confidence we have in them.

The paper convincingly describes how the new method is an advance over previous studies and is therefore likely to provide the most accurate 2000-year proxy SAM reconstruction to date. However, and the authors state this explicitly, it does not solve the problem of the marked disparity across the different reconstructions. Nevertheless, the work is a significant step in this field and I recommend it for publication subject to some (very) minor points listed below.

Minor points

Line 79: a recent study (Marshall et al. 2022: <https://doi.org/10.1007/s00382-022-06292-3>) also demonstrates the non-stationarity of SAM-SAT relationships during the instrumental record using reanalyses to show the non-stationarity is associated with spatial changes in SAM structure.

Line 138: should be Gong & Wang rather than Gong et al. (and similar for other citations of this reference in the manuscript).

The word 'index' is sometimes capitalised and sometimes not; e.g. even within the 'Reanalysis and Instrumental Indices' section.

Whether the SAM is positive or negative at a point in time is a function of the period chosen to normalize the index over. Using alternating periods of positive and negative SAM is an easy way to describe temporal SAM variability (e.g. line 189) but perhaps a comment stating a change in the sign of the SAM has no special physical meaning would be appropriate here.

The Marshall Index is (necessarily) based on the location of 12 points rather than the Gong and Wang definition used in the methodology. So, even using identical datasets you wouldn't expect a perfect match between the two definitions: another reason why the resultant correlation of 0.72 is actually rather good.

Line 243: double 'that'

Line 280: Given the authors recognise that one of the key issues in this field is the marked divergence between the different SAM reconstructions to date, I would like to have seen a little more detail regarding the causes of the discrepancies between them.

For example, they match each other relatively well from 1600-1750 but then diverge until the 20th Century: can this be linked directly to changes in proxy data availability or some other aspect of the different methodologies used?

Figure 5: make the A14 and V12 line colors more different

The software used to create the reference list has removed a lot of capital letters (e.g. for Antarctica, ENSO, SAM etc).

Reviewer #3 (Remarks to the Author):

King et al. reconstruct the Southern Annular Mode index over the common era using paleoclimate data assimilation. SAM reconstructions are important for understanding the largest mode of variability in the Southern Hemisphere, which affects climate in many regions of the Southern Hemisphere. Previous studies have used proxy records to reconstruct the SAM index over the last millennium, but data assimilation offers many advantages to the previous methods. Other studies have used DA to reconstruct the SAM index (which are not mentioned), but this study offers an advance to those by reconstructing longer timescales, including hydroclimate-sensitive proxy records, using a multi-model prior, and including an important optimal sensor analysis. Many studies have linked recent changes in the SAM to anthropogenic forcing, so reconstructing the SAM using DA on longer timescales than previous studies provides additional context for recent changes. Their reconstruction supports these previous findings, finding further evidence that recent SAM trends are a response to anthropogenic forcing, on multidecadal timescales. I commend the authors on this study which is well written, includes polished and digestible figures, and will provide the code used to generate the analyses – a practice more authors should follow.

Major concerns:

1. I have concerns about the way the study is presented, which leaves out two important reconstructions of the SAM that also use DA—Dalaiden et al., 2021 (which you've already cited as an example for sea ice) and O'Connor et al., 2021 (which uses a very similar method to yours). Although these two studies only include the last 100-200 years, leaving out these reconstructions from the study (1) overplays the advances of this study, making it seem like the first study to use DA to reconstruct the SAM Index (advances to the previous millennia-scale reconstructions are emphasized, but advances to previous DA-based SAM reconstructions are not mentioned), and (2) leaves out important comparisons that should be made. Given that DA is becoming increasingly common and that there are several paleoclimate DA frameworks, it is important to compare DA reconstructions of the same index/variable so that we can compare methods for different applications.
2. Given the significant drop off in proxy availability before 1500 and the need to adjust the variance to account for this, this is a major uncertainty in the reconstruction prior to 1500 that needs to be mentioned in the main text of the paper, including in the optimal sensor analysis section. Along these lines, some of the key caveats and limitations should also be mentioned in the main text, i.e. the discussion.
3. I am concerned that the skill metrics may be over-inflated as a result of forgetting to account for autocorrelation in the correlation calculations. This is standard practice for time series analysis. Additionally, trend analyses should be done using significance levels of 95% or greater. I'm not sure if the authors follow this or whether conclusions are made using 90%.

Line-by-line comments:

L123 – On the description of DA: the text describes a specific paleoclimate DA technique,

rather than defining DA in general. I would change the wording to reflect that (i.e., DA doesn't necessarily include proxy data). Furthermore, they are explaining a specific paleoclimate DA technique. DA doesn't necessarily include a forward model (i.e., Dalaiden et al. assimilation method).

L130 – These studies demonstrate the variety of variables that can be reconstructed with DA, but there are several studies that would be relevant to cite here, such as other DA reconstructions over the common era (i.e., sea reconstruction by Brennan and Hakim, 2022) and a reconstruction of Antarctic pressure and winds in recent centuries, including the SAM index (O'Connor et al., 2021, Dalaiden et al., 2021).

L222 – I am not following this section quite as well. I suggest revising the wording throughout this paragraph since it provides the evidence to back up the claim that recent trends are a response to anthropogenic forcing. Is this distribution of preindustrial trends shown somewhere? Is a trend considered significant if its p-value is <0.05 and its value is outside of the distribution of natural trends (or a confidence interval of natural trends)? A table could be helpful to explain the results and differences between the choice of time period used to define the preindustrial. There appear to be several trends that could be similar to trends near the 1980s (i.e., in the 1st century and near 1000). Are the trends in the 1980s statistically different from these previous trends? It could be worth noting the values of these trends for comparison of how much the trends have strengthened in recent decades relative to natural variability. The significance level used in this analysis needs to be clearly stated, as it appears that 90% and 95% confidence may be in the figure. Conclusions should only be made using at least 95% confidence.

Figure comments:

Figure 1 -- It would be more helpful to see the spatial correlations between the variables that are most reflective of the proxy data (temperature and precipitation) and the SAM Index, so that we can get a better idea for which proxies are in the most valuable regions (even better, using the covariance pattern in your climate model prior). Again, they need to account for autocorrelation in their calculations if they do not already.

Figure 4 – I would suggest only using 95% confidence, not 90%. Are the dotted lines 95%?

REVIEWER #1

King and colleagues have produced the first data-assimilation based reconstruction of the SAM over the Common Era. The reconstruction shows a positive trend over the 20th century, which the authors attribute to anthropogenic forcing. The DA reconstruction is compared to three others published over the past several years, and also is used to investigate which proxies contribute the most information to the reconstructed climate variability. This information is useful in the context of a sensor placement optimization exercise, and may inform further proxy development in the critically under-studied Southern Hemisphere.

This is a well-written paper worthy of publication in NComms for its novelty and findings, but I do have some comments that could strengthen the presentation and clarity of the results and conclusions:

Thank you for the recommendation! Below we describe in detail how we have addressed your specific comments.

1. I was craving more information about the proxy system models used in the main text. This can be very confusing, because the use of that phrase has very different meaning in the DA literature compared to the physically-based proxy system modeling literature (e.g. Evans, Dee et al.) – are the authors truly using a PSM, like VS Lite, or just a linear or bi-variate linear calibration with temperature and precipitation? At least one sentence being clear about this in the earlier sections of the text would be helpful.

Agreed - we have now added a line to the introduction indicating that we use seasonal linear temperature forward models for the PAGES-2k network, and the non-linear PDSI estimators for the drought atlases. (lines 148-150)

As the reviewer notes, ‘PSM’ in the paleoclimate data assimilation literature has come to encompass all of the forward operators that translate climate model output into the ‘space’ of the proxies for the comparison step. (including statistical models not based on an explicit “system”) But we agree that additional clarity is useful here, and as such, we now use “forward model” throughout the paper to refer to the statistical linear models, rather than ‘PSM’.

2. The use of multiple model priors is great, and should be emphasized in the main text.

Thank you! We now mention the multi-model prior in both the abstract (lines 27-28) and the introduction (lines 146-148). We have also now moved the “Caveats and limitations” section to the main text, which discusses some of the benefits of the multi-model prior (lines 419-435). Finally, we have added a paragraph comparing our reconstruction to two other DA SAM reconstructions, and we note the multi-model prior as a potential source of improved skill for our assimilation (lines 356-359).

3. It is fundamentally interesting that the reconstructions differ so much before the 20th century. I’m wondering if, beyond the speculative text from 271-288, if the authors could find a way to truly diagnose the drivers of the divergence in the reconstructions? Or prove / quantify the effects of “seasonal expressions” for example?

Agreed - and we definitely understand the interest in these discrepancies. At the moment, we can use our results for informed speculation, but we are hesitant to go much beyond what our results here indicate.

We do discuss the most likely drivers of divergence in the reconstructions in greater detail in the Introduction. Specifically, we provide more background for calibration with a non-stationary SAM index (lines 77-99), differing seasonal expressions (lines 101-106), and different proxy networks (lines 106-125) as causes of these differences. If the reviewer prefers, we can consider moving some of this text to the Discussion. For now though, we feel that these topics provide important context for the remainder of the paper, and thus do belong in the Introduction. For example, the issue of calibration with a non-stationary SAM index was one of our primary motivations for using DA (as it does not rely on such a calibration), and the issue of proxy networks provides the motivation for the optimal sensor analysis. However, we recognize that the reader may have forgotten about this previous text by the time they reach the Discussion, so we have reworded lines 336 to link the reader back to the Introductory text on this subject.

4. In the abstract and elsewhere, the authors spend time discussing how their reconstruction circumvents issues central to other studies like stationarity, but I feel these ideas are underdeveloped.

Thank you - We have reworded the abstract to provide more context before mentioning teleconnection stationarity (lines 26-31). We have also added a line to the introduction

explaining that our reconstruction circumvents this issue by relying on the calibration of proxy forward models to local climate variables, rather than the direct calibration of our proxy records to an instrumental SAM index (which then implicitly assumes a stationary teleconnection, lines 156-158). We also now refer the reader back to this text when mentioning calibration stationarity in the discussion (line 336).

5. The paper would benefit in the conclusions but perhaps also at the end of the abstract with. A statement on how the positive trend in the SAM might manifest in terms of climate changes, or compound with Southern Hemisphere climate change? For example, in which regions does the positive trend in the SAM cause drought, and what are the implications for Antarctic sea ice?

We have added a line to the end of the abstract stating that our results imply that the SAM's recent positive trend does result from anthropogenic climate change (lines 36-37). We also have added greater detail to the conclusion, where we note that this positive trend is linked to more severe droughts, more intense wildfires, reduced sea ice distribution, and increased ice shelf collapse. (lines 464-466)

25 variability and trends over what timescale?

Thank you for spotting this - we now specify that we are referring specifically to the Common Era (line 25)

30 this seems like a bit high-level for the abstract, and without context doesn't make sense here – the idea of stationarity between proxy records and SAM, perhaps phrase this more globally or move it to the discussion/results../ intro

We have reworded the abstract in order to provide the context for teleconnection non-stationarity before describing our own efforts. lines 26-31

34 implications of your findings? Last sentence should be implications – perhaps remove the sentence mentioned above at 30 and replace with this.

Thank you for the suggestion. We now write that the results of our trend analysis imply that the recent positive trend in the SAM is a result of anthropogenic climate change. (lines 36-37)

53-55 what about Antarctic sea ice, ice shelf collapse, mitigation of warming around E. Antarctica, climate change compounding impacts?

Thank you for these suggestions. We now mention and have added references for these effects (lines 54-56)

72 I'm still wary of this very broad use of the words "trends and variability" – in what fields, and on what time scales, and how do these features indicate "large discrepancies?"

Thank you for this observation - we have reworded this to indicate that we are referring to a lack of decadal to centennial coherence in the SAM index prior to the 1850s. We also refer the reader to Hessel et al., 2017 for further discussion. (Lines 75-77)

123 Not exactly recent, 2016 is already 6 years ago and the original Gosse papers came out well before that!

Agreed - We removed "recently developed"

138 This sentence seems out of place and perhaps belongs in the methods, or in a different section.

We have added several sentences before this one that now provide an overview of our method (lines 144-152) to set the stage and place it in context – for example, the forward models used and our multi-model ensemble. As these new sentences are also more methodological, we feel that this sentence is now more in keeping with the surrounding lines.

179-180 I'm a little confused by this on first read, can you be more explicit about what you mean when you say "our reconstruction is not calibrated directly to the SAM index." You are reconstructing the SAM index, . . .

Agreed - we have reworded the introduction (lines 156-158) to clarify that our method does not calibrate the proxy records directly on a SAM index. Instead, our method

relies first and foremost on a calibration of the proxy forward models with local climate variables. We have also added “As discussed in the introduction” on line 336 so as to refer the reader back to this section.

202 why? (why does the connection to stratosphere/tropical variability etc.) lead you to say the uncertainty is overestimated?

We have clarified that it is the DA method’s lack of incorporation of inter-annual persistence (which in the real world and models would derived from connections to the stratosphere/tropical variability) that cause the uncertainty to be overestimated. Lines 234-236

206 sentence structure here personifies “proxy” a bit too much for my style preferences, I would say “the potential for any one proxy record to reduce...”

Agreed - we have reworded the sentence as suggested. Line 241

217 use of “common” here is a bit vague, do you mean consistent?

Agreed - we have changed the line to state “consistent” instead of “common”. Line 256-261

218 Figure 4 is quite difficult to interpret without more information in the main text

Thank you - we have added several lines to the main text explaining how (1) the wavelet coherence plot is lacking any band of statistical significance (lines 250-255), (2) the composite-mean response in the superposed epoch analysis is not statistically significant (line 255-261), and (3) Additional explanation of how to read and interpret the trend plots (lines 263-272). We have also added several lines to the Figure 4 caption explaining how to interpret the figures.

233 rephrase? = “A significant trend in the...”

Corrected the typo and also reworded slightly.

235 rephrase “We also experimented”

Changed to “experimented”

236 similar how?

Clarified that we are referring to the very recent and longer-term trends being outside the range of natural variability diagnosed from the reconstruction (line 291-292)

268 ozone-depleting substances – I would spell out what those are like CFCs etc. and be specific

Thank you for the suggestion - we now refer to chlorofluorocarbons (line 324).

320-321 I was craving more information about the proxy system models used in the main text. This can be very confusing, because the use of that phrase has very different meaning in the DA literature compared to the physically-based proxy system modeling literature (e.g. Evans, Dee et al.) – are the authors truly using a PSM, like VS Lite, or just a linear or bi-variate linear calibration with temperature and precipitation? At least one sentence being clear about this in the earlier sections of the text would be helpful.

Agreed - we have now added a line to the introduction indicating that we use seasonal linear temperature forward models for the PAGES-2k network, and the non-linear PDSI estimators for the drought atlases. (lines 148-150)

As the reviewer notes, ‘PSM’ in the paleoclimate data assimilation literature has come to encompass all of the forward operators that translate climate model output into the ‘space’ of the proxies for the comparison step. (including statistical models not based on an explicit “system”) But we agree that additional clarity is useful here, and as such, we now use “forward model” throughout the paper to refer to the statistical linear models, rather than ‘PSM’.

Based on Section starting in methods 489, it appears you are simply using linear regression. But the main text suggests dependencies on both moisture and temperature (?)

The non-linear PDSI estimators used for the drought atlases depend on both moisture and temperature, as per Equation 5. To reduce confusion, we have added a line to the beginning of this section that indicates that we use two different types of forward models (prior to describing any specific model). Lines 610-611

340 suggest starting new sentence “Such assessments..”

Agreed - changed as suggested to begin a new sentence.

341 might be worth mentioning the explicit effects, eg. Drought, fire, etc. here again – this is important.

Thank you for the suggestion - we now explicitly mention a number of specific climate effects in the conclusion (lines 464-466).

342 it is fundamentally interesting that the reconstructions differ so much before the 20th century. I’m wondering if, beyond the speculative text from 271-288, if the authors could find a way to truly diagnose the drivers of the divergence in the reconstructions? Or prove / quantify the effects of “seasonal expressions” for example?

We definitely understand the interest in these discrepancies, and would similarly like to know their exact causes.

We do now discuss the most likely drivers of divergence in the reconstructions in greater detail in the Introduction. Specifically, we provide more background for calibration with a non-stationary SAM index (lines 77-99), differing seasonal expressions (lines 101-106), and different proxy networks (lines 106-122) as causes of these differences. If the reviewer prefers, we can consider moving some of this text to the Discussion. For now though we feel that these topics provide important context for the remainder of the paper, and thus do belong in the Introduction. For example, the issue of calibration with a non-stationary

SAM index was one of our prime motivations for using DA (as it does not rely on such a calibration), and the issue of proxy networks provides the motivation for the optimal sensor analysis. However, we recognize that the reader may have forgotten about this previous text by the time they reach the Discussion. so we have reworded lines 336 to link the reader back to the Introductory text on this subject.

349- can you close with a statement on how the positive trend in the SAM might manifest in terms of climate changes, or compound with Southern Hemisphere climate change? For example, in which regions does the positive trend in the SAM cause drought, and what are the implications for Antarctic sea ice?

Agreed - We have added a line to the end of the abstract stating that our results imply that the SAM's recent positive trend results from anthropogenic climate change (lines 36-37). We also have added greater detail to the conclusion, where we note that this positive trend is linked to more severe droughts, more intense wildfires, reduced sea ice distribution, and increased ice shelf collapse. (lines 464-466)

Supplemental Figure 1, what are the units of the trend??

The units of the trend have been normalized such that the mean and variance of the detrended reconstruction match the mean and variance of the detrended Marshall index over the period 1958-2000. To clarify, we have labeled Supplemental Figure 1b with "Normalized Trend". We have also provided a detailed description of the normalization in the Figure 4 caption, and refer the reader to this process in the caption of Supplemental Figure 1.

Supplemental Table S3 the total number of proxy records used in the reconstruction should be mentioned in the main text. Also Figure 2 doesn't seem to jive with Table S3, perhaps because of the addition of the SADA and ANZDA data?

Yes, the SADA and ANZDA data cause the differences between Figure 2 and Table S3. Figure 2 indicates the total number of assimilated records, which includes data from PAGES2k, SADA, and ANZDA. By contrast, Table S3 summarizes of the PAGES2k records used in the assimilation. Following the reviewer's suggestion, we now indicate the total number of assimilated PAGES2k records (N=40) in the main text (line 144).

All figures should be labeled with panels e.g. a, b, c, d for the caption clarity.

Agreed - we have added such labels to the Figures.

FIGURE TODO: Add subpanel labels

Figure 4 caption/main text needs more explanation on how to read these figures.

Thank you for this feedback. We have now added several lines to both the main text (lines 251-272) and the caption explaining how to interpret these figures.

REVIEWER #2

The paper is well-written in a clear and logical style. The methods and materials section is exhaustive and should enable the work to be reproduced. The noteworthy component of the paper is that it uses a new (to this field) proxy data assimilation technique to compute a 2000-year proxy index of the Southern Annular Mode (SAM). This is an important but relatively new area of research in Southern Hemisphere climatological studies. Previous SAM reconstructions of similar length are quite different from each other, which significantly reduces the confidence we have in them.

The paper convincingly describes how the new method is an advance over previous studies and is therefore likely to provide the most accurate 2000-year proxy SAM reconstruction to date. However, and the authors state this explicitly, it does not solve the problem of the marked disparity across the different reconstructions. Nevertheless, the work is a significant step in this field and I recommend it for publication subject to some (very) minor points listed below.

Thank you! We have provided details about how we addressed the reviewer's specific points below.

Minor points

Line 79: a recent study (Marshall et al. 2022: <https://doi.org/10.1007/s00382-022-06292-3>) also demonstrates the non-stationarity of SAM-SAT relationships during the

instrumental record using reanalyses to show the non-stationarity is associated with spatial changes in SAM structure.

Thank you for this suggestion and pointing us toward this reference. We have now added a reference to Marshall et al. 2022, as well as noted that the non-stationarity is associated with changes to the SAM's spatial structure. Lines 91-93.

Line 138: should be Gong & Wang rather than Gong et al. (and similar for other citations of this reference in the manuscript).

Thank you - yes, changed to Gong and Wang throughout the manuscript (lines 151, 210, 471, 487)

The word 'index' is sometimes capitalised and sometimes not; e.g. even within the 'Reanalysis and Instrumental Indices' section.

Thank you, agreed - changed to use lowercase when discussing the SAM, Marshall, and Fogt indices.

Whether the SAM is positive or negative at a point in time is a function of the period chosen to normalize the index over. Using alternating periods of positive and negative SAM is an easy way to describe temporal SAM variability (e.g. line 189) but perhaps a comment stating a change in the sign of the SAM has no special physical meaning would be appropriate here.

Thank you for this observation, which is an important point of clarification. We have now added two lines explaining that a change in sign has no particular physical meaning (lines 213-216).

The Marshall Index is (necessarily) based on the location of 12 points rather than the Gong and Wang definition used in the methodology. So, even using identical datasets you wouldn't expect a perfect match between the two definitions: another reason why the resultant correlation of 0.72 is actually rather good.

Thank you. We have now added a line noting that the indices differ by definition and thus we would not expect perfect agreement amongst them. Lines 209-211

Line 243: double 'that'

Thank you - fixed

Line 280: Given the authors recognise that one of the key issues in this field is the marked divergence between the different SAM reconstructions to date, I would like to have seen a little more detail regarding the causes of the discrepancies between them. For example, they match each other relatively well from 1600-1750 but then diverge until the 20th Century: can this be linked directly to changes in proxy data availability or some other aspect of the different methodologies used?

Thank you for the interest. We note that we do discuss potential causes for these discrepancies at length in the introduction. Specifically, we suggest calibration with a non-stationary SAM index (lines 77-99), differing seasonal expressions (lines 101-106), and different proxy networks (lines 106-125) as causes of these differences.

However, we hesitate to discuss “direct” causes of these differences. We definitely understand the interest in these discrepancies, and would similarly like to know their exact causes. However, knowing the exact causes of the reconstruction’s differences would require a deep dive into the specific techniques and methodological choices used by these existing reconstructions, which is well beyond the scope of this paper.

If the reviewer prefers, we could move some of the introductory text to the discussion, but we feel that this text provides important context for the remainder of the paper. For example, the issue of calibration with a non-stationary SAM index was our motivation for using DA (as it does not rely on such a calibration), and the issue of proxy networks provides the motivation for the optimal sensor analysis. However, we recognize that the reader may have forgotten about this previous text by the time they reach the discussion. As such, we have reworded line 336 to help remind and link the reader back to the introductory text.

Figure 5: make the A14 and V12 line colors more different

Agreed - we have updated the figure to use more different colors. Specifically, we have made A14 a somewhat darker blue, and have given V12 a slightly purple tint.

The software used to create the reference list has removed a lot of capital letters (e.g. for Antarctica, ENSO, SAM etc).

Thank you for catching this (this was a LaTeX issue) - we have corrected the reference list to retain capital letters.

REVIEWER #3

King et al. reconstruct the Southern Annular Mode index over the common era using paleoclimate data assimilation. SAM reconstructions are important for understanding the largest mode of variability in the Southern Hemisphere, which affects climate in many regions of the Southern Hemisphere. Previous studies have used proxy records to reconstruct the SAM index over the last millennium, but data assimilation offers many advantages to the previous methods. Other studies have used DA to reconstruct the SAM index (which are not mentioned), but this study offers an advance to those by reconstructing longer timescales, including hydroclimate-sensitive proxy records, using a multi-model prior, and including an important optimal sensor analysis. Many studies have linked recent changes in the SAM to anthropogenic forcing, so reconstructing the SAM using DA on longer timescales than previous studies provides additional context for recent changes. Their reconstruction supports these previous findings, finding further evidence that recent SAM trends are a response to anthropogenic forcing, on multidecadal timescales. I commend the authors on this study which is well written, includes polished and digestible figures, and will provide the code used to generate the analyses — a practice more authors should follow.

Thank you for the commendation! Below we provide the details of how we have attempted to address the reviewer's specific concerns.

Major concerns:

1. I have concerns about the way the study is presented, which leaves out two important reconstructions of the SAM that also use DA—Dalaiden et al., 2021 (which you've already cited as an example for sea ice) and O'Connor et al., 2021 (which uses a very similar method to yours). Although these two studies only include the last 100-200 years, leaving out these reconstructions from the study (1) overplays the advances of this study, making it seem like the first study to use DA to reconstruct the SAM Index (advances to the

previous millennia-scale reconstructions are emphasized, but advances to previous DA-based SAM reconstructions are not mentioned), and (2) leaves out important comparisons that should be made. Given that DA is becoming increasingly common and that there are several paleoclimate DA frameworks, it is important to compare DA reconstructions of the same index/variable so that we can compare methods for different applications.

Thank you for the suggestions! We now reference these two additional reconstructions in the introduction (lines 139-143). We also have added the correlation coefficients of these reconstructions with the Marshall index to the results section (lines 199-201). We have also added a paragraph discussing the differences in correlation coefficients for all three DA-generated SAM indices (lines 351-362). We note that the correlations of these two particular products are somewhat lower than ours, and we also discuss some methodological differences. Specifically, we note that Dalaiden et al. uses a particle filter (in contrast to the Kalman filter of this study and O'Connor et al) and discuss the use of single-model versus multi-model ensembles. Finally, we note the increase in hydroclimate information provided by the drought atlases, which provide additional skill in our reconstruction.

2. Given the significant drop off in proxy availability before 1500 and the need to adjust the variance to account for this, this is a major uncertainty in the reconstruction prior to 1500 that needs to be mentioned in the main text of the paper, including in the optimal sensor analysis section. Along these lines, some of the key caveats and limitations should also be mentioned in the main text, i.e. the discussion.

Thank you for the feedback. We have added a paragraph to the main text noting the use of the variance correction and explaining its justification. (lines 180-190)

Regarding the optimal sensor analysis section, there are two types of variance discussed in the paper, and it is possible to confuse these: We discuss both (1) the temporal variance of the reconstructed time-series, and (2) the variance of the posterior ensembles. The variance correction is used to adjust for artifacts in the former, whereas the optimal sensor analysis is based on the quantification of the latter. In the framework of our ensemble square-root Kalman filter, these two types of variance are determined independently of one another. (Variance 1 derives from the posterior mean, whereas variance 2 derives from the posterior deviations). As such, the variance correction scheme does not affect the results of the optimal sensor analysis.

Because of the potential for confusion, though, we now elect not to mention the variance correction scheme in the discussion of the optimal sensor (since this presents the largest risk of confusing these two things). To reduce this potential confusion, we now refer to the optimal sensor analysis variance using variants of “reconstruction uncertainty” or “constrained reconstruction uncertainty” (and so avoid the word “variance”) within the main text. We take the same approach within the Methods, excluding a single line used to connect the idea of reconstruction uncertainty to variance across the posterior (lines 670-671).

Separately, in response to the reviewer’s final point, we have moved the “Caveats and Limitations” section into the main text of the discussion. (lines 400-444)

3. I am concerned that the skill metrics may be over-inflated as a result of forgetting to account for autocorrelation in the correlation calculations. This is standard practice for time series analysis. Additionally, trend analyses should be done using significance levels of 95% or greater. I’m not sure if the authors follow this or whether conclusions are made using 90%.

Thank you for the note. We adjusted our correlation calculations to account for reduced sample size resulting from AR1 autocorrelation, following the method outlined by Bretherton et al., 1999. We have noted this step in the main text (lines 203-204) and added a reference to Bretherton et al., 1999 in the bibliography. However, our p-values still remain at very low values – less than $1E-4$ - so we have not altered the “ $p_{ij} < 0.001$ ” values in the main text.

Our trend analyses were previously conducted at a 90% level. Following the reviewer’s suggestion, we have updated our results and Figure 4 to use a 95% confidence interval instead. Qualitatively, we find that the results for the 95% confidence interval are not markedly different from the previous results for 90%, and so we have only minimally altered the text. The main change is on line 287, where we have changed the interval from 55-80 years to 60-80 years.

Line-by-line comments:

L123 – On the description of DA: the text describes a specific paleoclimate DA technique, rather than defining DA in general. I would change the wording to reflect that (i.e., DA doesn’t necessarily include proxy data). Furthermore, they are explaining a specific paleoclimate DA technique. DA doesn’t necessarily include a forward model (i.e., Dalaiden et al. assimilation method).

Thank you - We have reworded this to indicate that we refer to the case when DA is used as a reconstruction technique (as opposed to analyses like optimal sensors that don't require proxy data). Line 128

However, we have a different perspective on whether paleoclimate DA requires a forward model. It's true that, in some cases, it's possible to compute a proxy estimate directly from climate model output fields. However, we would argue that this represents the case of an "identity" forward model, in which a proxy record is implicitly assumed to perfectly record the climate variable described by a climate model grid point. Following the framework of Evans et al., 2013 - an identity forward model indicates that (1) the proxy record is treated as a perfect sensor of (spatially-averaged) local climate, (2) the proxy record is not subject to archival effects, and (3) the proxy record is not subject to observational effects. And these choices are all important methodological distinctions.

Regarding the assimilation method in Dalaiden et al. (2021) "Reconstructing atmospheric circulation and sea-ice extent in the West Antarctic over the past 200 years using data assimilation", their Section 2.4 does explicitly describe the proxy system models used in the experiment, with Equations 4 and 5 detailing the (statistical) forward models used to estimate tree ring widths from climate model output.

L130 – These studies demonstrate the variety of variables that can be reconstructed with DA, but there are several studies that would be relevant to cite here, such as other DA reconstructions over the common era (i.e., sea reconstruction by Brennan and Hakim, 2022) and a reconstruction of Antarctic pressure and winds in recent centuries, including the SAM index (O'Connor et al., 2021, Dalaiden et al., 2021).

Thank you for the suggestions. We have added references to these papers (lines 136-140)

L222 – I am not following this section quite as well. I suggest revising the wording throughout this paragraph since it provides the evidence to back up the claim that recent trends are a response to anthropogenic forcing.

Thank you for the feedback. We have added several lines explaining how significance is assessed (lines 262-272), and we have also added several lines describing how to interpret Figure 4cd.

Is this distribution of preindustrial trends shown somewhere?

We have elected not to display the trend distributions because we feel that plotting these values would hinder the clarity of Figure 4. Because we calculate a distribution for each different trend length, we have (71 distributions * 2 time intervals) for Figure 4d alone, and we feel that adding 142 data points (dimensions) would unnecessarily complicate the figure. However, we have reworded the text to make sure it is clear that we use many different distributions here, rather than a single one (lines 267-272).

Is a trend considered significant if its p-value is ≤ 0.05 and its value is outside of the distribution of natural trends (or a confidence interval of natural trends)?

Yes, and we have added a line explicitly stating our use of the 95% confidence level (line 270)

A table could be helpful to explain the results and differences between the choice of time period used to define the preindustrial. There appear to be several trends that could be similar to trends near the 1980s (i.e., in the 1st century and near 1000). Are the trends in the 1980s statistically different from these previous trends? It could be worth noting the values of these trends for comparison of how much the trends have strengthened in recent decades relative to natural variability.

The differences between the two time periods are illustrated by the dashed and dotted lines in Figure 4cd. The trends in the 1980s are statistically different from the 1st and 9th century trends when enclosed by the dotted lines, which show results when using the interval 1-1900 CE to establish significance thresholds. We have rewritten this section to help clarify how significance is being assessed, and that the dashed and dotted lines display results for different time intervals, rather than different confidence levels or another quantity. (lines 263-272)

The significance level used in this analysis needs to be clearly stated, as it appears that 90% and 95% confidence may be in the figure. Conclusions should only be made using at least 95% confidence.

We now explicitly state the significance level of 95% on line 270, and have updated Figure 4 to use 95% levels, rather than 90%.

Figure 1 – It would be more helpful to see the spatial correlations between the variables that are most reflective of the proxy data (temperature and precipitation) and the SAM Index, so that we can get a better idea for which proxies are in the most valuable regions (even better, using the covariance pattern in your climate model prior).

Thank you for the suggestion. We definitely understand the interest in other ways of visualizing which proxies are in the most valuable regions (and we previously considered this exact suggestion when drafting the paper). However, the complication we found in showing these correlation maps is that the individual proxies are associated with a wide range of different specific seasonal response windows, and we rely on these different windows in the DA framework. For example, we might estimate proxy *A* using climate variable averages from DJF, but proxy *B* is estimated using averages over JJASON, and so on. So in order to illustrate the correlations of all the proxies with the SAM via correlation maps, we would need to produce a map for each unique season and variable. Ultimately, we use 27 seasons x 2 variables in this study, which are difficult to display in a single plot. One approach we considered was to use lots of panels for ‘small multiples’ maps (a la Pat Bartlein), but in this case the scale of the domain and the number of proxies and proxy types would not result in a very readable figure either. It is for this reason we have elected to instead show the DJF sea level pressure field. Because this is the field used to derive the SAM index, it has a single season of interest within the DA, as well as a large spatial expression. We also note that the proxies are taken from the PAGES2k dataset (all temperature sensitive) and the Drought Atlases (all soil moisture data). We also think that the results from the optimal sensor analysis (which identify the proxies in the most valuable locations) shown in Figure 3ab are useful in this regard, as these figures do account for the differing seasonal and climate variable responses of the proxies. The optimal sensor analysis additionally incorporates proxy uncertainties (e.g. which proxies have a high signal-to-noise ratio with local climate and which have weaker relationships) into the results, which would not be apparent in a ‘small multiples’ correlation figure.

However, if the reviewer and editor would prefer it, we could include additional maps in the Supplemental Material.

Again, they need to account for autocorrelation in their calculations if they do not already.

Agreed - We now use the method outlined by Bretherton et al., (1999) to account for autocorrelation in our calculations.

Figure 4 – I would suggest only using 95% confidence, not 90%. Are the dotted lines 95%?

Both the solid and dotted lines show significance at the same level. The solid lines use the interval 1500-1900 CE to determine the significance threshold, whereas the dotted lines use the interval 1-1900 CE. We have added several lines to the main text to help explain this more clearly (lines 263-272), and we have also reworded the caption to make this more clear. In the original figure, these lines displayed values for the 90% confidence level, but we have updated the figure to use the 95% confidence level and now state the level in the main text (line 270). We have also reworded the caption slightly to help make this more clear.

References

Reviewer #1 (Remarks to the Author):

I have reviewed the response to reviewer file and checked on the authors' revision. I find this revision addresses all of my major comments and remains convincing and well-written. I have no further comments on the manuscript and support its publication when the comments of the other reviewers have been adequately addressed.

Reviewer #3 (Remarks to the Author):

I appreciate the authors' thoughtful responses and that they took many of the suggestions from the reviewers. It is an improvement over the last draft, but I have two outstanding concerns:

1. I would like further clarification on the statements about the significance of the modern trends relative to the Common Era and its attribution to anthropogenic forcing. The authors state that the modern trend is outside the range of natural variability on time intervals of ~40-55 years long, which especially exceptional for its persistence. In Figure 5 I see a long positive trend from at least ~1000-1150 which appears to be of comparable (if not greater) magnitude than that of the late 20th century (shown in both the reconstruction presented in this study and in D18). Can the authors comment on the magnitude of this trend and why the 20th century trend is outside the range of natural variability, even when including this time period? From eyeballing figure 5, it appears that the change from ~1100 to 1150 is ~2, and the change from 1950-2000 is ~1.5. If the claim that recent trends are outside the range of natural variability is true even though a similar or stronger trend occurred ~1100, I think that is important to note that the trend is not unprecedented in the Common Era.

2. I appreciate the authors' addition of the comparison to the D21 and O21 reconstructions. However, the statistics the authors appear to be taken from the D21 and O21 studies, which is problematic because the exact methods for calculating the statistics may not be identical (esp. the significance level), but more importantly the time periods are not identical. The calculation only requires downloading the reconstruction data from two studies and repeating simple analyses the authors have already computed. The authors suggest that their reconstruction is more skillful due to the use of a multi-model prior, but the statistics need to be properly calculated to defend such statements. The comparison is very important to make given that this study emphasizes the importance of using proxy DA. This study makes many improvements to the D21 and O21 studies, but a proper comparison needs to be made to highlight the benefits of using the multi-model prior and the additional proxy records. The paleo DA community would appreciate this! I also want to note that the O21 study is improperly cited: the study is published in GRL. I have listed the citations for D21 and O21 so that it is absolutely clear which studies I am referring to:

Dalaiden, Q., Goosse, H., Rezsöhazy, J., and Thomas, E. R.: Reconstructing atmospheric circulation and sea ice extent in the West Antarctic over the past 200 years using data assimilation, *Climate Dynamics*, 57, 3479–3503, 10.1007/s00382-021-05879-6, 2021.

O'Connor, G. K., Steig, E. J., and Hakim, G. J.: Strengthening Southern Hemisphere westerlies and Amundsen Sea Low deepening over the 20th century revealed by proxy-data assimilation, *Geophysical Research Letters*, 48, e2021GL095999, 10.1029/2021GL095999, 2021.

REVIEWER #3

I appreciate the authors' thoughtful responses and that they took many of the suggestions from the reviewers. It is an improvement over the last draft, but I have two outstanding concerns:

Thank you for the continued feedback! It has certainly helped improve the clarity and scope of this paper. Below we describe in detail how we have addressed your remaining concerns.

1. I would like further clarification on the statements about the significance of the modern trends relative to the Common Era and its attribution to anthropogenic forcing. The authors state that the modern trend is outside the range of natural variability on time intervals of 40-55 years long, which is especially exceptional for its persistence. In Figure 5 I see a long positive trend from at least 1000-1150 which appears to be of comparable (if not greater) magnitude than that of the late 20th century (shown in both the reconstruction presented in this study and in D18). Can the authors comment on the magnitude of this trend and why the 20th century trend is outside the range of natural variability, even when including this time period? From eyeballing figure 5, it appears that the change from 1100 to 1150 is 2, and the change from 1950-2000 is 1.5. If the claim that recent trends are outside the range of natural variability is true even though a similar or stronger trend occurred 1100, I think that is important to note that the trend is not unprecedented in the Common Era.

Thank you for raising this point of clarification, which we address here as well as with modifications to the manuscript. When we refer to the modern trend being “outside the range of natural variability”, we mean that the modern trend is outside of the 95% CI of the distribution of trends (as assessed from the reconstruction). We do not intend to indicate that the modern trends are greater than all previous trends. We recognize now that the previous text was ambiguous, and so we have added text throughout the manuscript indicating that the trend is outside of the 95% CI, rather than greater than all other trends. (Lines 269, 276, 278, 317, 321, 327, 481-482)

Regarding the feature specifically near 1150: we first note that we assess the 95% CI of trends using two different background periods. The first period (1500-1900 CE; solid lines in Figure 4cd), does not include the feature at 1150 by design. Thus, the solid contours in Figure 4cd do not take the 1150 feature into consideration. That said, we do

also assess the 95% CI of trends using the period 1-1900 CE (dashed lines in Figure 4cd), which *does* include 1150. Notably, the significance of the modern trend is much reduced when using this background (i.e. when taking the trends near 1150 into account).

We use these two different periods to assess the 95% CI in an attempt to balance considerations of reconstruction length with the greater uncertainty of the reconstruction further back in time. We use 1500-1900 as a background period because it is the interval in which the drought atlases are informing the reconstruction. This interval has the greatest amount of available proxy information, and so the 1500-1900 background represents the smallest reconstruction uncertainty (at the cost of a shorter length over which to assess the range of behavior of the SAM). By contrast, the 1-1900 background uses the reconstruction for as long as possible, but at the cost of higher reconstruction uncertainty. For example, the 1150 feature has substantially greater uncertainty than any feature in the interval 1500-1900, making us cautious of judging the mean tendency of this feature compared to the more accurate reconstructed trends in more recent centuries. Ultimately, the choice of background interval reflects a tradeoff between length and uncertainty, and we feel these dual intervals help provide some balance.

To help communicate this, we have added several lines that observe the 1150 feature, comment on the greater uncertainties during the time period, and examine the feature in the context of other reconstructions and climate forcings (lines 292-300).

2. I appreciate the authors' addition of the comparison to the D21 and O21 reconstructions. However, the statistics the authors appear to be taken from the D21 and O21 studies, which is problematic because the exact methods for calculating the statistics may not be identical (esp. the significance level), but more importantly the time periods are not identical. The calculation only requires downloading the reconstruction data from two studies and repeating simple analyses the authors have already computed. The authors suggest that their reconstruction is more skillful due to the use of a multi-model prior, but the statistics need to be properly calculated to defend such statements. The comparison is very important to make given that this study emphasizes the importance of using proxy DA. This study makes many improvements to the D21 and O21 studies, but a proper comparison needs to be made to highlight the benefits of using the multi-model prior and the additional proxy records. The paleo DA community would appreciate this!

At that Reviewer's request, we have now downloaded and re-calculated the statistics as suggested. The correlations (and p-values) we report for the reconstructions in D21 and O21 now use the same time interval and statistical approach as the correlations reported for our own reconstruction. Specifically, correlations are assessed against the

DJF Marshall index over the period 1958-2000, and p-values are adjusted to account for temporal autocorrelation following the method outlined by Bretherton et al., 1999. We initially report these statistics on lines 200-201, and then provide a greater discussion of the statistics (and call the attention of the reader to the fact that we have recalculated them) on lines 359-366.

Overall, the results are similar as before, in that the D21 and O21 reconstructions correlate with the Marshall index at lower levels than our reconstruction. With the exception of some minor changes for clarity, we have therefore not changed the subsequent discussion of the three DA products (lines 366-377).

I also want to note that the O21 study is improperly cited: the study is published in GRL. I have listed the citations for D21 and O21 so that it is absolutely clear which studies I am referring to:

Dalaiden, Q., Goosse, H., Rezsöhazy, J., and Thomas, E. R.: Reconstructing atmospheric circulation and sea ice extent in the West Antarctic over the past 200 years using data assimilation, *Climate Dynamics*, 57, 3479–3503, [10.1007/s00382-021-05879-6](https://doi.org/10.1007/s00382-021-05879-6), 2021.

O'Connor, G. K., Steig, E. J., and Hakim, G. J.: Strengthening Southern Hemisphere westerlies and Amundsen Sea Low deepening over the 20th century revealed by proxy-data assimilation, *Geophysical Research Letters*, 48, e2021GL095999, [10.1029/2021GL095999](https://doi.org/10.1029/2021GL095999), 2021.

Our apologies - and thank you for the clarification. We have corrected and updated the reference list accordingly. (Lines 988-991)